# Recapitulation of Structure–Function–Regulation of Blood–Brain Barrier under (Patho)Physiological Conditions

**DOI:** 10.3390/cells13030260

**Published:** 2024-01-30

**Authors:** Hin Fong, Botao Zhou, Haixiao Feng, Chuoying Luo, Boren Bai, John Zhang, Yuechun Wang

**Affiliations:** 1Faculty of Medicine, International School, Jinan University, Guangzhou 510632, China; mekof@stu2021.jnu.edu.cn (H.F.); lawjudycheukwing@stu2021.jnu.edu.cn (C.L.); bai0127@stu2021.jnu.edu.cn (B.B.); 2Department of Physiology, Basic Medical and Public Health School, Jinan University, Guangzhou 510632, China; zbt101@stu2020.jnu.edu.cn; 3Gies College of Business, University of Illinois Urbana-Champaign, Urbana-Champaign, IL 61801, USA; haixiao4@illinois.edu; 4Department of Physiology and Pharmacology, Loma Linda University, Loma Linda, CA 92350, USA; jhzhang@llu.edu; 5Department of Neurosurgery, Loma Linda University, Loma Linda, CA 92350, USA

**Keywords:** blood–brain barrier, integrity, regulation, health, diseases

## Abstract

The blood–brain barrier (BBB) is a remarkable and intricate barrier that controls the exchange of molecules between the bloodstream and the brain. Its role in maintaining the stability of the central nervous system cannot be overstated. Over the years, advancements in neuroscience and technology have enabled us to delve into the cellular and molecular components of the BBB, as well as its regulation. Yet, there is a scarcity of comprehensive reviews that follow a logical framework of structure–function–regulation, particularly focusing on the nuances of BBB regulation under both normal and pathological conditions. This review sets out to address this gap by taking a historical perspective on the discovery of the BBB and highlighting the major observations that led to its recognition as a distinct brain barrier. It explores the intricate cellular elements contributing to the formation of the BBB, including endothelial cells, pericytes, astrocytes, and neurons, emphasizing their collective role in upholding the integrity and functionality of the BBB. Furthermore, the review delves into the dynamic regulation of the BBB in physiological states, encompassing neural, humoral, and auto-regulatory mechanisms. By shedding light on these regulatory processes, a deeper understanding of the BBB’s response to various physiological cues emerges. This review also investigates the disruption of the BBB integrity under diverse pathological conditions, such as ischemia, infection, and toxin exposure. It elucidates the underlying mechanisms that contribute to BBB dysfunction and explores potential therapeutic strategies that aim to restore the BBB integrity and function. Overall, this recapitulation provides valuable insights into the structure, functions, and regulation of the BBB. By integrating historical perspectives, cellular elements, regulatory mechanisms, and pathological implications, this review contributes to a more comprehensive understanding of the BBB and paves the way for future research and therapeutic interventions.

## 1. Introduction

It is widely recognized that the central nervous system (CNS) serves as one of the control systems regulating the activities of all organs and systems, and the brain communicates with other organ systems in the body to ensure homeostasis for the survival of the total organism.

Considering the brain’s paramount importance, numerous protective mechanisms have evolved to safeguard it. Normally, the concept of a “brain barrier” encompasses three defenses: the blood–brain barrier (BBB), the blood–cerebral-spinal-fluid (CSF) barrier, and the CSF–brain barrier. In addition, the intramural periarterial drainage pathway [1] and glymphatic system [2] have been found to also play important roles in the maintenance of the stability of the CNS. Among all the protective mechanisms, the BBB stands out as the most significant and well understood. The BBB is a highly specialized structure that acts as a selective barrier, controlling the exchange of substances between the blood and the brain. It regulates the internal ion concentration within the brain and the passage of neurotransmitters, macromolecules, and neurotoxins into the brain.

The microstructure of the BBB and the complex mechanisms governing its permeability have far-reaching implications for various physiological and pathological conditions. Over the decades, eminent researchers have endeavored to unravel the complexities of the BBB, including its dynamic nature, structural intricacies, and role in the development of numerous neurological diseases, ranging from neurodegenerative disorders like Alzheimer’s disease to inflammatory conditions such as multiple sclerosis. In addition to mechanism studies, more researchers are interested in uncovering innovative therapeutic approaches for the treatment of CNS disorders. 

The following sections aim to provide a comprehensive survey of the BBB’s structure and functions, tracing its historical discovery and delving into the present understanding of its regulation in both health and diseases. This paper endeavors to narratively and visually introduce how the BBB was established and has evolved from a spatial–temporal dimension, covering the basic structure, functions, and regulation of the BBB under healthy and diseased conditions, with a focus on the research advances in neuroscience.

## 2. Structure and Functions of BBB under Healthy Condition

### 2.1. The Discovery of the Barrier between Circulating Blood and the Brain

The BBB has been a topic of intense research interest since its postulation in the late 19th century. Paul Ehrlich, using tissue-staining techniques, observed that certain dyes injected into the peripheral circulation permeated almost all tissues but not the brain or CSF. Ehrlich therefore proposed that the brain’s high lipid content prevented dye affinity [3]. Goldmann, a student of Ehrlich, conducted further research in 1909 and 1913 [4,5]. He found that injecting the dye trypan blue into the brain ventricles resulted in the staining of the brain tissue, while intravenous administration did not stain the brain or spinal cord. He then hypothesized that the CSF, entering the brain tissue via the choroid plexuses, acted as a transporter for certain substances. These observations and hypotheses contributed to the understanding of the BBB and its role in regulating the passage of substances between the bloodstream and the brain.

### 2.2. The Proposal of the Term BBB 

The dynamics of the substance transfer between the peripheral blood, CSF, and brain were extensively studied by pioneering neurophysiologist Lena Stern and her team in the early 20th century [6]. Their findings suggest that the interface between the brain and the rest of the body is semipermeable and operates bidirectionally, with the BBB effectively limiting the permeability of some circulating substances into the brain while facilitating the permeability of compounds delivered to the brain into the bloodstream [7]. These findings led Stern and her colleagues to develop the concept of the BBB, which posits that CECs play a dual role in the defense and support of the CNS [8]. This proposition was an important advance in our understanding of the BBB. By the late 1920s, the existence of the BBB was widely recognized [8]. 

Stern et al. first coined the term BBB [9]. They conducted detailed studies on the penetration of a wide range of molecules from the blood into the CSF, brain, and within the subarachnoid and ventricular CSFs of adult animals. Stern was the first to highlight the importance of the brain’s interstitial fluid environment for its normal function, aligning with Claude Bernard’s milieu interne [10].

Further studies showed that substances with the ability to penetrate the brain can also be found in the CSF. In addition, it has been found that substances can enter the brain through the CSF without having to cross the BBB [7]. The above observations suggest that the CSF plays a crucial intermediary role in the transportation of blood compounds into the brain. This finding led to the development of a preliminary mathematical model to understand the molecular diffusion between blood and CSF at the cellular level [11]. In the late 1960s, Reese and Karnovsky finally determined that the BBB mainly consists of astrocyte endfeet and capillary endothelia using electron microscopic cytochemical studies [8]. 

### 2.3. The Microstructure of the BBB

The advent of the transmission electron microscope in 1967 was a pivotal moment in the study of the microstructure of the BBB. By the late 1960s, studies had pinpointed the exact location of the mammalian BBB in the endothelia of cerebral capillaries [12]. These investigations demonstrated that the barrier, which comprises the plasma membrane and cell body of ECs, as well as the tight junctions (TJs) between adjacent cells, prevents macromolecular tracers like horseradish peroxidase (HRP) from passing beyond the first luminal interendothelial TJs. Further observations highlighted that, in the choroid plexus, HRP was restrained from moving from the blood to the CSF by TJs connecting the apical regions of the epithelial cells [13]. The role of the endothelium in the BBB was confirmed via freeze-fracture studies, which showed that the TJs between the ECs of CNS capillaries and venules are arranged in parallel [14,15].

Unlike peripheral ECs, CECs are characterized by the absence of fenestrations and the presence of TJs. Further studies using tracers and microelectrodes revealed their low permeability and high electrical resistance, which underlie their protective role [16]. The complex network formed by these cell–cell junctions was unveiled through freeze-fracture studies conducted by Farquhar and Palade [17]. This research identified specific components, including claudins, occludins, adhesion molecules, and various adhesion-junction proteins, that are integral to the molecular structure and barrier integrity of the BBB [18]. Following the first discovery of occludins [19], claudins were identified [20,21], with claudin-5 emerging as a major component of brain capillary TJs [22,23].

It is now well known that ECs in the BBB create a highly selective barrier that allows essential nutrients to flow from the bloodstream to the brain while inhibiting potentially harmful substances (Figure 1) [24]. For example, polarized transporters on the ECs mediate the tightly regulated process of vital nutrient and waste transfer throughout the BBB. Solute transporters, like GLUT-1, are mostly found on the luminal side of the canaliculus and aid in the CNS’s absorption of critical nutrients. Conversely, efflux transporters are situated on the luminal side of the EC and provide robust protection against the intrusion of potentially harmful chemicals into the CNS (Figure 1C) [25]. 

### 2.4. Study on Neighboring Cells Involved in BBB 

Electron microscopy provided detailed insights into the ultrastructure of blood vessels, revealing that not only endothelial cells but also the BM and surrounding astrocytic endfeet contribute to the barrier function [26,27,28,29]. Extensive research has revealed that the distinctive nature of the BBB is the result of intricate interactions between ECs and neighboring cells, including pericytes, astrocytes, and neurons [7,8] (Figure 2). 

The NVU is a complex functional unit composed of both the nervous component and vascular component, mainly including neurons, glial cells, endothelial cells, pericytes, the basement membrane, and the extracellular matrix. The BBB functions as part of the NVU [30] and plays an important role in maintaining cerebral vascular function and structural stability, and it is an important barrier between the blood and brain tissue. Transendothelial fluid transfer, the BBB integrity, and neurovascular coupling are all maintained via the collective action of NVU cells [31]. The following are descriptions related to the surrounding cellular components involved in the regulation of the BBB. 

#### 2.4.1. Pericytes 

As far back as the late 19th century, the researchers Eberth and Rouget first identified pericytes as individual cells with distinctive morphological characteristics located on the exteriors of capillaries [32,33]. Pericytes cover between 22% and 99% of the endothelial cell surface [34]. Their distribution varies, with a higher proportion of circumferential protrusions at the arterial end of the capillary bed and longitudinal protrusions more prevalent in the middle section. At the venous end of capillaries, pericytes exhibit a stellate or star-like morphology [35]. Pericytes are engaged in a range of functions, including matrix regeneration, angiogenesis, the regulation of EC proliferation, and the promotion of neural stem cell properties. These functions highlight their significant potential in the treatment of CNS injuries and other conditions [36]. Moreover, pericytes are contractile cells that are separately located on capillaries. They have been observed to manage the physiological flow of blood in the brain. Following an ischemic event, these cells reduce the blood flow by causing the capillaries to constrict before they perish, and they are implicated in the regulation of the local blood pressure [37].

#### 2.4.2. Astrocytes

Astrocytes constitute another pivotal component of the BBB. Astrocyte terminal processes, known as endfeet, play a critical role in modulating the neuronal activity and cerebral blood flow through close interactions with brain blood vessels and neuronal synapses [38]. These endfeet express specialized molecules that regulate BBB protein transporters and ion concentrations [7]. Furthermore, astrocytes form functional synapses through gap junctions, facilitating coordinated responses [39]. Additionally, they regulate enzymatic systems, facilitate tight-junction formation, and control transporter polarization through the release of growth factors such as GDNF, bFGF, vascular endothelial growth factor (VEGF), and ANG-1 [40].

#### 2.4.3. Microglia 

As resident immune cells of the CNS, microglia have a significant impact on the integrity of the BBB. They are involved in a wide range of functions and demonstrate a remarkable level of heterogeneity in their phenotypes. This heterogeneity is particularly noticeable when considering the various subsets of microglia that have been identified. These subsets, distinguished based on their phenotypic characteristics, receptor expressions, and functional roles, contribute to the complex and dynamic functions of the brain’s immune system [41].

In the healthy brain, homeostatic microglia are the dominant subset. These microglia maintain normal brain function and surveillance, demonstrating a distinct gene signature that includes genes such as P2ry12, Tmem119, and Hexb [42]. Conversely, reactive microglia become activated in response to injury, infection, or neurodegenerative diseases. These cells undergo morphological changes and upregulate inflammatory markers, contributing to the brain’s immune response to various challenges [43]. Disease-associated microglia are characterized by a distinct gene expression profile, including upregulated genes involved in phagocytosis and lipid metabolism [44]. Lastly, disease-specific microglial subsets have been identified in various conditions, such as multiple sclerosis. These subsets express specific markers, including CD11c and MHC-II, and may contribute to the disease’s progression or the brain’s response to the disease [42]. Notably, the CD11c marker is conventionally associated with dendritic cells and macrophages. However, it is also expressed by a subset of microglia, particularly in the context of Alzheimer’s disease and experimental autoimmune encephalomyelitis [42].

#### 2.4.4. Interactions among BBB and Neighboring Cells

Astrocyte endfeet, in conjunction with pericytes embedded in the BM, provide structural support for brain CECs. The intricate connections between astrocytes, endothelial cells, and pericytes are upheld by the basement membrane, which consists of a complex interplay between laminin, nidogen, heparan sulfate, and collagen IV, forming a three-dimensional protein structure [45]. Neuropathological disorders may lead to significant alterations in the molecular composition of the BM. Additionally, astrocytes regulate the formation and maintenance of the endothelial BBB properties. Understanding the complex interactions between microglia and the BBB is pivotal for developing effective strategies to enhance BBB restoration in various pathological conditions.

## 3. Regulation of BBB’s Integrity under Normal Condition

Based on the source, the regulatory factors of the BBB integrity include both endogenous and exogenous factors. The endogenous factors normally refer to the factors originating from within the body, including neurotransmitters, hormones, and immune factors [46]. The exogenous factors are from the external environment outside the body, such as chemical and physical factors [47]. Moreover, in terms of the time course and spatial distribution, spatial and temporal regulation play crucial roles. These categories often intersect and overlap to a certain extent. From a physiological perspective, there are essentially four types of regulatory mechanisms: neural regulation, humoral regulation, immunoregulation, and auto-regulation (Figure 3). In this section, we begin by introducing the classical regulatory patterns under physiological conditions and then explore the common factors that impact the BBB integrity through these regulatory patterns. 

### 3.1. Neural Regulation of BBB

Neural regulation entails physiological processes directly influenced by the nervous system. Neural regulation plays a pivotal role in the development of the BBB, especially during the early embryonic stages. Brain vascular development commences with the pre-formed perineural vascular plexus encircling the nascent neural tube [48]. As vascular development progresses, VEGF, expressed and released by neural progenitor cells within the avascular neuroepithelium, stimulates the inward expansion of capillaries from the pre-formed perineural vascular plexus [49]. The interaction between VEGF/VEGFR and the Dll/Notch pathway is vital for the subsequent stem cell differentiation processes [50]. Neural activities have also been shown to regulate developmental CNS angiogenesis, cerebral hemodynamics, and the cells of the NVU [51,52]. During the development, a fundamental barrier arises in the endothelia of growing vessel sprouts. This close connection between angiogenesis and barrier formation suggests that the signals inducing barrier development likely originate from neural progenitor cells. In the neural regulation of the BBB, key neurotransmitters, such as glutamate, may directly impact the BBB, as the presence of active ionotropic glutamate receptors in ECs has been observed in in vivo experiments [53]. Moreover, the BBB permeability increases after the direct application of 1 mM of glutamate [54]. Neural activity can further enhance the local cerebral blood flow through vasodilation [51], a phenomenon known as neurovascular coupling [55]. For example, activating channelrhodopsin specifically expressed in excitatory neurons elevated the local blood flow in anesthetized rodents [56]. Additionally, excitatory neurons produce prostaglandin E2 through the expression of cyclooxygenase-2, and prostaglandin E2 induces vasodilation by binding to the EP2 and EP4 prostanoid receptors on smooth muscle cells [57]. 

### 3.2. Humoral Regulation of BBB

Humoral regulation involves the production and secretion of special chemicals (such as hormones, metabolites, etc.) by certain cells. These chemicals are transported to target cells through bodily fluids and regulate the activities of these cells by interacting with corresponding receptors on the cell’s surface. Hormonal regulation is the primary form of humoral regulation. The BBB does not completely impede the entry of hormones into the brain, and the constituent cells of the BBB are significantly influenced by hormones [58]. The following is a list of several common hormones that play a regulatory role in regulating the BBB’s permeability. 

Estrogen can influence the permeability of the BBB by binding to sites on tight junctions [59]. For example, 17β-estradiol inhibits the activation of nuclear factor-κB-dependent *matrix metalloproteinase (MMP) genes* on the *MMPs’ gene* promoters, thereby inhibiting *MMP* transcription and mitigating tight-junction disruption [60]. Furthermore, there is substantial evidence indicating the protective role of testosterone in the BBB integrity [61]. One study suggests that male *mice* with chronic testosterone deficiency may experience heightened BBB permeability by disrupting the TJ architecture and decreasing the level of claudin-5 and ZO-1. However, the BBB permeability and TJ integrity of the castrated mice were restored through testosterone supplementation [62].

Glucocorticoids have a therapeutic effect on BBB disruption [63]. For example, hydrocortisone can induce the expression of occludin and enhance the barrier function of the BBB by activating the glucocorticoid receptor, which binds to putative glucocorticoid response elements in the occludin promoter [64]. On the contrary, high doses of dexamethasone (5–20 μM) increase cytotoxicity and elevate the monolayer cell permeability of brain ECs [65].

Insulin can influence multiple aspects of the brain EC function, such as modulating transporters for amino acids, leptin, and p-glycoprotein [58]. It also influences the alkaline phosphatase and glutamate-cysteine ligase catalytic subunits, maintaining the intracellular level of glutathione, which could protect the integrity of the BBB indirectly [66].

### 3.3. Auto-Regulation of BBB 

Auto-regulation pertains to the ability of some organs, tissues, and cells to adapt to changes in the surrounding environment without relying on neural and hormonal regulation. In the context of the BBB, auto-regulation primarily refers to factors that directly affect the BBB components without neural or humoral regulation. For example, the BBB can be directly influenced by certain chemical and physical factors, such as temperature and pH, as reviewed by Liu, W.Y. et al. [63]. One study suggests that even minor fluctuations in the physiological pH range may affect the BBB’s control over paracellular permeation by modulating the claudin-5 expression [67]. The expression of the TJ protein decreased under the condition of ultrasound, resulting in an increase in the permeability of the BBB [68]. The permeability of the BBB also increased under the condition of electromagnetic pulse explosion [69], and shear stress is vital for inducing and maintaining the BBB [70]. Research has demonstrated that shear stress upregulates the expressions of ZO-1 and occludin, enhancing the function of the brain microvascular barrier [63]. A reduction in the shear force can lead to the disruption of the BBB [71]. Brief heat shocks can affect the BBB functions, and at higher temperatures, it takes more time for the BBB to recover. Furthermore, the repeated application of heat treatment can induce heat resistance in brain microvascular ECs [72]. 

### 3.4. Factors That Affect the BBB under Normal Conditions

Under normal conditions, various common factors, like circadian rhythms, diet, and intestinal microbiota, can affect the integrity of the BBB either individually or in combination with the regulation patterns discussed earlier.

#### 3.4.1. Circadian Rhythm 

The circadian rhythm is an endogenous oscillatory biological process that can synchronize with external factors like light, temperature, and feeding patterns but is not entirely dependent on external clues [73]. As reviewed by Nicolette S. and Michal T. in *mammals*, the molecular-clock core regulates circadian rhythmic oscillations through a transcription–translation feedback loop, and this complex signaling mechanism ultimately influences the BBB integrity by modulating the oscillation of TJPs and the transporter functionality [74]. For example, elevated Mg^2+^ levels in daytime subperineurial glia, which are maintained via the molecular clock in perineurial glia cells, enhance the activity of P-glycoprotein transporters, leading to a reduction in the BBB permeability [75]. Furthermore, BBB efflux transporter proteins also exhibit circadian oscillations in both *flies* and *rodents* [76,77]. 

#### 3.4.2. Psychological Stress Factors

Exposure to different sources of stress can activate physiological responses to maintain the internal balance, and this adaptation to psychological stress can impact the brain in various ways [78,79]. During psychological stress, the activation of the hypothalamic–pituitary–adrenal axis leads to the secretion of corticotropin-releasing hormone, which subsequently stimulates the release of adrenocorticotropic hormone. This cascade culminates in the synthesis of glucocorticoids, as summarized in the review paper by Charmandari E., Tsigos C., and Chrousos G. [80]. Prolonged exposure to high levels of cortisol can potentially harm brain health [81]. Current research suggests a link between psychological stress and BBB functions. For example, in a stress state, hypothalamic–pituitary–adrenal-axis activation and elevated cortisol levels induce the mobilization of bone marrow-derived monocytes [82], which are recruited to the brain by activated microglial cells [83]. This leads to neuroinflammatory responses and changes in the brain’s vascular system [84]. Moreover, psychological stress has pro-inflammatory effects mediated by mast cell activation [85] and is associated with BBB opening [86].

#### 3.4.3. Diet and Nutrients 

Neurons in the CNS require essential nutrients across the BBB to maintain normal functions [87]. For example, glucose, fatty acids, and amino acids are transported across the BBB through carrier-mediated transport or receptor-mediated transcytosis. Additionally, vitamins are actively transported to maintain their brain levels.

In animal models, malnutrition and nutrient imbalances have been associated with potential BBB damage. However, these effects are often linked to inflammatory responses, making it challenging to determine the direct impact of malnutrition and nutrient imbalances on the BBB itself. For instance, in Kanosky’s study, *rats* fed a high-energy diet exhibited reduced mRNA expressions of the TJPs Claudin-5 and Claudin-12 in the choroid plexus. Furthermore, the expressions of occludin, Claudin-5, and Claudin-12 in the BBB microvasculature were decreased [88]. Additionally, some studies have suggested that caffeine may regulate the BBB by blocking adenosine receptors, inhibiting cAMP phosphodiesterase activity, or modulating intracellular calcium ions, helping to stabilize the integrity of the BBB under various neurodegenerative disease conditions [89]. Although the exact mechanisms remain unclear, these findings provide insights into the potential implications of everyday human dietary habits.

#### 3.4.4. Intestinal Microbiota

Intestinal microbiota are reported to play a certain role in maintaining BBB functions [90]. Research has shown that the absence of intestinal microbiota significantly reduces the expressions of occludin and claudin-5 [91]. Intestinal microbiota can convert complex carbohydrates into short-chain fatty acids, with various impacts. For instance, butyric acid could strengthen the BBB by tightening the connections between cells [92], and sodium butyrate increases the TJ protein expression, thereby reducing the permeability [93]. A comprehensive categorization of the types of intestinal microbiota that influence the BBB can be found in the review paper by Parker, A.; Fonseca, S.; Carding, S.R. [94].

#### 3.4.5. Exercise and Environmental Exploration 

Research suggests that heightened neural activity resulting from exercise could elevate the brain levels of Insulin-like Growth Factor 1 [95], which is secreted by the liver and binds to Insulin-like Growth Factor 1 receptors that are abundant in CECs. Inhibiting neural activity using tetrodotoxin can prevent Insulin-like Growth Factor 1 accumulation [96]. Additionally, exercise can significantly improve the expression of brain-derived neurotrophic factor, a neurotrophin, in CECs damaged by hypertension [97]. Studies have also demonstrated that exploring new environments and whisker stimulation have exercise-like effects, and relocating *mice* from dark to well-lit conditions leads to significant changes in the transcription of BBB-related genes in the visual cortex [98]. 

## 4. Blood–Brain Barrier under Disease Conditions

The BBB acts as a guardian of the brain, maintaining homeostasis within the CNS. However, the BBB’s permeability is not a fixed attribute and can be modulated by various harmful agents encountered in our daily lives. In this section, we first describe the primary factors that disrupt the BBB and then provide an example involving heavy metals to explain how these elements can lead to diseases. 

### 4.1. Types of BBB Damage and Underlying Mechanisms 

Several factors can impact the BBB integrity through different mechanisms, leading to various diseases. The primary factors contributing to BBB disruption under diseased conditions are summarized in Table 1, which includes the following: (1) mechanical damage resulting from trauma or surgical operations; (2) ischemia caused by a reduced blood flow to the brain tissue, leading to BBB damage, neuronal injury, and cell death; (3) the infiltration of immune cells during inflammation, where immune cells, such as T cells and monocytes/macrophages, can infiltrate the brain tissue and contribute to BBB disruption; (4) the activation of MMPs, enzymes that degrade the BBB’s extracellular matrix, leading to increased permeability and BBB disruption; (5) oxidative stress induced by the increased production of reactive oxygen species (ROS), causing BBB damage and potentially cell death; (6) various chemicals and biological factors (details in Table 1).

All of these factors can affect the main components of the BBB, including the cells and supporting structures, like the basement membrane, resulting in various diseases.

Here, we take heavy metals as an example to show how they affect the BBB integrity and lead to increased permeability and the disruption of its regulation (Table 2).

#### 4.1.1. Direct Disruption of Tight Junctions

Heavy metals, such as lead, mercury, cadmium, and arsenic, can directly target the proteins involved in forming tight junctions between ECs in the BBB [114,115,116], as reviewed by Kim J-H [114,115,116]. These metals can disrupt the structure and function of the TJPs, leading to a breakdown in the integrity of the BBB. 

Zona occludens-2, a critical component of the zona occludens protein family, plays a pivotal role in the structural organization of TJs across the epithelial and endothelial cell layers. These proteins are essential for anchoring the transmembrane protein complexes to the cytoskeleton made of actin, thereby facilitating the proper placement and stabilization of the intercellular junctions. The integrity of the BBB is partly reliant on the proper functioning of zona occludens-2 [117]. Research findings have shown that, when exposed to lead, there is a significant reduction in the concentration of zona occludens-2, with the levels diminishing by 25%. This suggests that lead exposure directly undermines the TJ integrity.

#### 4.1.2. Oxidative Stress

Heavy metals can generate ROS and impair the antioxidant defense systems within the brain [115]. One example is cadmium exposure, which has been linked to a decrease in the microvessel antioxidant potential. Cadmium can increase the permeability of the BBB and raise the concentration of malondialdehyde in brain microvessels. Malondialdehyde is a marker of oxidative stress, suggesting that cadmium toxicity involves oxidative damage. 

The excessive production of ROS leads to oxidative damage to the BBB, including lipid peroxidation, protein oxidation, and DNA damage [116]. This oxidative stress can further increase the BBB permeability and compromise the regulation of the BBB.

#### 4.1.3. Inflammation

Heavy metals trigger an inflammatory response in the brain. They can activate immune cells and stimulate the release of pro-inflammatory cytokines and chemokines [118]. This inflammatory response can disrupt the normal functioning of the BBB by altering the expression of TJPs, increasing the permeability of the barrier, and facilitating the entry of immune cells into the brain.

#### 4.1.4. Metalloproteinase Activation

Heavy metals can activate MMPs, which are enzymes involved in the remodeling of the extracellular matrix. The excessive activation of MMPs can lead to the degradation of the basement membrane and extracellular matrix components of the BBB [119]. This degradation compromises the structural integrity of the barrier, resulting in increased permeability. The role of MMP-2/9 in the Pb-induced damage of the BBB is not known, but lead might trigger the disruption of the BBB by prompting astrocytes to release MMP-2/9. These enzymes, when induced by lead, could break down the TJ proteins occludin and ZO-1 found between neighboring endothelial cells. As a result, the structural integrity of the barrier could be compromised.

#### 4.1.5. Disruption of Transporters

Some heavy metals interfere with the function of the transporters present at the BBB. For example, mercury has been shown to inhibit the activity of nutrient transporters, such as glucose transporters and amino acid transporters, leading to impaired nutrient exchange and cellular homeostasis in the brain [120]. These disruptions can affect the normal regulation of essential substances across the BBB.

The cumulative effects of these mechanisms result in a disruption in the BBB regulation, leading to increased permeability. This heightened permeability allows heavy metals and other harmful substances to cross the BBB and enter the brain. This can result in neurotoxic effects, including neuronal damage, inflammation, oxidative stress, and various neurological disorders. Understanding these effects holds potential for the development of therapeutic strategies for neurological diseases in which BBB dysfunction contributes to disease progression, such as multiple sclerosis and stroke. Therefore, the study of the interactions between these chemicals and the BBB has important implications for our understanding of environmental factors in neurological health. Further research in this field can pave the way for novel therapeutic strategies and preventive measures for a broad spectrum of neurological conditions.

**Table 2 cells-13-00260-t002:** Summary of factors that affect the BBB permeability.

Toxic Factors	Mechanisms of Increased BBB Permeability	BBB Model	Dosage	Representative Reference
Ethanol	By increasing oxidative stresses	In vitro (human stem cells)	50 mM	Stoffel, R. D., Bell, K. T. and Canfield, S. G. (2020) [121].
Lead	By damaging tight junctions	In vivo (*mice*)	7-day exposure to Pb at 54 mg/kg and 4 weeks	Gu, H., Territo, P. R., Persohn, S. A., et al. (2020) [122].
Mercury	By increasing oxidative stresses	In vitro (*porcine*)	3 µM of organic mercury compounds and 100 µM of inorganic HgCl_2_	Lohren, H., Bornhorst, J., Fitkau, R., et al. (2016) [123].
Carbon monoxide	By damaging endothelial cell function and tight junctions	In vivo (*rats*)	Absorption of 2.5–3.0 mL/L CO for 1 h	Wang, X., Tie, X., Zhang, J., Wan, J. and Liu, Y. (2004) [124].
Cocaine	By downregulating TJPs, altering the expression of adhesion molecules, and promoting neuroinflammation	In vitro *(rat*)	5–20 mg/kg, ip	Barr JK, Brailoiu GC, Abood ME, et al. (2020) [125].
Nicotine	By altering tight-junction protein distribution and promoting inflammation	In vivo (*rat*)	4.5 mg free base per kilogram of body weight per day	Hawkins, B. T., Abbruscato, T. J., et al. (2004) [126].
PM 2.5 ^1^	By downregulating tight junctions and disrupting tight junctions in the BBB	In vitro	10 ug/mL	Kang, Y. J., Tan, H. Y., Lee, C. Y. and Cho, H. (2021) [127].
Pesticides ^2^	By inducing oxidative stress and inflammation and disrupting tight junctions	In vivo (*rat*)	1/50th of the LD50 for the pesticides quinalphos, cypermethrin, and lindane.	Gupta, A., Agarwal, R. and Shukla, G. S. (1999) [128].

Note: ^1^ PM 2.5: atmospheric particulate matter (PM) with a diameter of less than 2.5 μm. ^2^ Pesticides: organophosphate, cypermethrin, and lindane.

### 4.2. Restoration of BBB under Disease Conditions

The restoration of the BBB can occur naturally as part of the body’s healing process, but this recovery is often incomplete or insufficient, especially after severe insults like ischemic stroke. The endogenous repair mechanisms primarily involve the cells that constitute the BBB itself, such as endothelial cells, astrocytes, and pericytes. Using ischemic stroke as an example, the following details the cellular restoration of the BBB. 

#### 4.2.1. Endothelial Cells 

Under normal conditions, ECs can proliferate and realign to seal gaps in the barrier. However, during a stroke, ECs of the BBB undergo irreversible damage due to increased oxidative stress and inflammation, leading to changes in the TJPs. Autophagy, a cellular process for the degradation and recycling of damaged organelles and proteins, may play a beneficial role in ameliorating the breakdown of the BBB and loss of TJs after ischemia [129]. Although the effects of autophagy in ECs are likely to be context-dependent, enhancing autophagy’s ability is reported to aid in the recovery from a stroke, as ECs repair themselves through atugophagy process [130]. In the case of intracerebral hemorrhage, thrombin is clinically used. When injected, it injures the CECs and astrocytes, which are critical for maintaining the functions of the BBB. But after the initial injury, the brain initiates a repair process. New brain microvascular ECs and astrocytes start to form around the damaged vessels. This is the body’s natural response to heal and restore the integrity of the BBB [131]. Also, endothelial progenitor cells hold great promise as potential treatment elements in conditions such as acute ischemic stroke due to their ability to transform into mature endothelial cells. However, the scarcity of EPCs in peripheral blood presents a significant challenge, hindering their extraction and utilization for therapeutic purposes.

Moreover, recent studies have found that the recombinant human hepatocyte growth factor can counteract the reduction in the occludin and ZO-1 protein expressions in ECs following enduring cerebral ischemia. These proteins are crucial TJs that preserve the integrity and selective permeability of the BBB. According to the study by Guo et al. (2021b), during the acute stage of traumatic brain injury, recombinant human fibroblast growth factor 20 could upregulate the expressions of TJ proteins and adherens-junction proteins and also promote angiogenesis, thereby alleviating BBB damage [132]. All these mechanisms and methods help restore the permeability of the BBB to the normal condition.

#### 4.2.2. Pericytes 

Pericytes exhibit various repair mechanisms after BBB injury. Firstly, they have immunomodulatory properties and can regulate the inflammatory responses within the CNS. During acute insults, such as inflammation or injury, pericytes can modulate the activation and polarization of microglia, the immune cells of the CNS. This modulation helps maintain the balance between pro-inflammatory and anti-inflammatory responses, reducing the harmful effects on the BBB integrity [133]. Secondly, it has been discovered that pericytes and astrocytes communicate through tunneling nanotubes in response to astrocyte apoptosis triggered by ischemia or reperfusion injury. The formation of tunneling nanotubes between pericytes and astrocytes may serve as a call-for-help mechanism, allowing pericytes to transfer functional mitochondria to ischemic or apoptotic astrocytes [134]. This intercellular transfer of mitochondria could potentially rescue astrocytes from apoptosis and contribute to their survival. This process may contribute to both astrocytes’ survival and the restoration of BBB functions. Thirdly, pericytes have the ability to differentiate into other cell types, including endothelial cells. This property allows them to participate in angiogenesis, the formation of new blood vessels, and vasculogenesis, the generation of new endothelial cells. Studies suggest that the rapid accumulation of pericytes is observed in the peri-infarct area, and that they are involved in angiogenesis and BBB repair after stroke [135]. Also, pericytes can differentiate into neural- and vascular-lineage cells after ischemia, contributing to tissue repair and regeneration [136]. 

Finally, pericytes increase in number and enhance the secretion of the extracellular matrix (ECM) after a stroke, resulting in the formation of discrete fibrotic scars that are distinct from glial scars. PDGFR signaling is implicated in triggering fibrosis by promoting pericyte proliferation, differentiation into fibroblast-like cells, and the secretion of CM substrates. The full implications of this transformation are not yet fully understood but could potentially contribute to tissue repair and regeneration following a stroke [69,137]. 

#### 4.2.3. Astrocytes 

Astrocytes play a vital role in the BBB and the neurovascular unit, actively regulating BBB functions. They release various factors influencing the tight junctions between ECs forming the BBB. Some astrocyte-derived factors produced after injuries worsen BBB structure damage, leading to endothelial cell apoptosis and decreased tight-junction protein expression. For example, astrocytes are a source of MMPs that degrade tight junctions and cause the disruption of the BBB and the extracellular matrix after ischemia. However, some astrocyte-derived factors protect ECs and facilitate the rebuilding of TJPs [74]. For instance, post-ischemic neurons can stimulate astrocytes to produce VEGF, which supports BBB repair. Additionally, astrocytes regulate the water flux between the blood and the brain via aquaporin 4, potentially participating in BBB repair. Recent studies reveal that the lipogenesis of astrocytes is activated in the peri-infarct area in the subacute phase after cerebral ischemia injury, and that IL-33 plays a role in promoting lipogenesis [138]. Therefore, controlling various astrocyte-derived factors, such as retinoic acid, VEGF-A, and pentraxin-3, is an important strategy to repair the BBB.

#### 4.2.4. Microglia

Microglia, as the resident immune cells of the CNS, fulfill diverse roles in response to ischemic stroke. They contribute significantly to the development of the cerebral and retinal vasculatures, influencing the brain’s ability to respond to ischemic injury effectively. Upon activation, microglia produce cytokines and chemokines that regulate inflammation and the BBB’s permeability. Microglia damage the function by producing various inflammatory mediators, including MMPs and pro-inflammatory cytokines, which disrupt TJPs such as ZO-1 and claudin-5 [80]. Additionally, activated microglia phagocytose cellular debris, clearing damaged areas and suppressing inflammatory responses. Depending on the stimulus, microglia can adopt different functional phenotypes, including pro-inflammatory and anti-inflammatory states, both of which impact the BBB integrity and the overall tissue response to ischemia. Following brain injury or disease, microglia undergo activation and manifest distinct phenotypes known as M1 and M2. The M1 phenotype signifies a pro-inflammatory state, while the M2 phenotype is associated with an anti-inflammatory response. Activation of the M2 microglial phenotype exerts a protective and regenerative effect on the BBB. M2 microglia secrete various protective factors that mitigate inflammatory responses and facilitate BBB regeneration, thereby restoring the BBB integrity and function after disease or injury. Thus, increasing the ratio of M2 to M1 in the middle and late stages of inflammation is undoubtedly a good way to facilitate the BBB’s recovery. 

### 4.3. Therapeutic Strategies Targeted to BBB Integrity under Disease Condition

Endothelial cells, pericytes, astrocytes, microglia, and various chemical mediators intricately respond to ischemic stroke, influencing outcomes like the BBB integrity, inflammation, and tissue remodeling. Despite their roles in potential repair processes, these innate mechanisms often fall short, especially after severe damage. This is where external interventions come in. Importantly, understanding the complex interactions among these cells that constitute the BBB is crucial for developing effective strategies to enhance the BBB restoration in various pathological conditions.

One key aspect of the BBB integrity lies in the regulation of TJPs. TJP loss escalates the BBB permeability, enabling harmful substances to enter the brain, contributing to neurological disorders. Recent research focuses on modulating TJPs to enhance drug delivery to the brain. Strategies include using cytokines, like transforming growth factor-beta (TGF-β), to elevate the claudin-5 and occludin expressions [99]. Chemical mediators such as neuroactive steroids maintain the BBB integrity, exemplified by 17b-estradiol and progesterone. Targeting the BBB cellular components and their interactions also presents promising therapeutic avenues. Controlling astrocyte-derived factors like retinoic acid and VEGF-A aid BBB repair. Effectively mitigating the detrimental effects of microglial activation on the BBB in various conditions is crucial.

It is worth mentioning that some physical factors and nanoparticles hold potential for enhancing drug delivery by increasing the BBB permeability. For example, under specific ultrasound conditions, the expression of BBB TJPs decreases, and they undergo redistribution, resulting in transient BBB dysfunction. The duration of this effect varies depending on the ultrasound treatment conditions [68,139]. In addition, exposure to electromagnetic pulse treatment can also lead to an increased BBB permeability, possibly due to changes in the localization and decreased expression of the TJ protein ZO-1 [69]. Moreover, nanoparticles that target tight junctions can disrupt the integrity, allowing therapeutic-agent passage [80].

In summary, the therapeutic strategies aim to enhance the body’s innate repair mechanisms or introduce additional mechanisms to speed up recovery and ensure a more complete restoration of the BBB. These strategies can include anti-inflammatory treatments, drugs that stabilize or enhance the functions of the BBB’s cellular components, and even cell therapies that introduce new cells to replace those damaged or lost during injury.

## 5. Conclusions and Perspective

In this comprehensive exploration of the BBB, we have delved into the intricate balance that this vital protective interface maintains under both healthy and diseased conditions. The BBB, under normal circumstances, exhibits a remarkable ability to regulate the passage of molecules, safeguarding the CNS. This dynamic interplay involves intricate humoral and neural mechanisms, with factors such as hormones and circadian rhythms influencing the BBB permeability. However, in the face of adversity, the vulnerabilities of the BBB become evident. Detrimental agents can disrupt the BBB through mechanisms like oxidative stress, inflammation, and direct damage to tight junctions, leading to increased permeability and neurotoxic effects, ultimately resulting in the development of neurological disorders.

Our exploration reveals the innate restorative mechanisms orchestrated by the cellular components of the BBB, where endothelial cells, pericytes, astrocytes, and microglia form a formidable defense line against disease and damage. Yet, these natural repair processes often fall short in the face of severe damage. It is at this crucial juncture where external interventions must play a pivotal role. Signs have indicated a significant disparity in the incidence of cerebrovascular disease across various racial groups. Could this variation be linked to differences in the structure and functions of the BBB among different ethnic populations? This intriguing question warrants further investigation.

Recent years have witnessed increased attention on different in vitro BBB models, alongside the development of corresponding techniques to measure the BBB permeability. In the age of big data and artificial intelligence, advanced technologies are anticipated to enhance our understanding of the BBB structure, functions, and regulation. These advancements are likely to lead to significant breakthroughs in the prevention and treatment of related diseases. This endeavor is not merely a scientific quest but is also a path toward enhancing the resilience of our CNSs, opening new horizons for neurological health and well-being.

## Figures and Tables

**Figure 1 cells-13-00260-f001:**
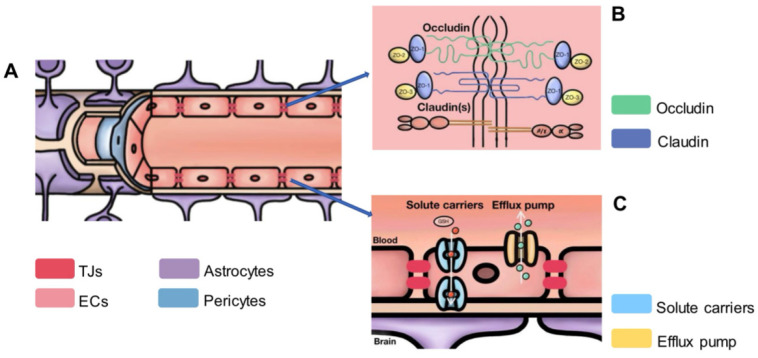
The BBB and its detailed molecular components: (**A**) longitudinal profile of the BBB; (**B**) the TJPs, including occludin and claudin, contribute to the integrity of the BBB; (**C**) transport mechanisms, such as solute carriers and efflux pumps, actively regulate the passage of substances, emphasizing the BBB’s vital role in maintaining the brain’s stability.

**Figure 2 cells-13-00260-f002:**
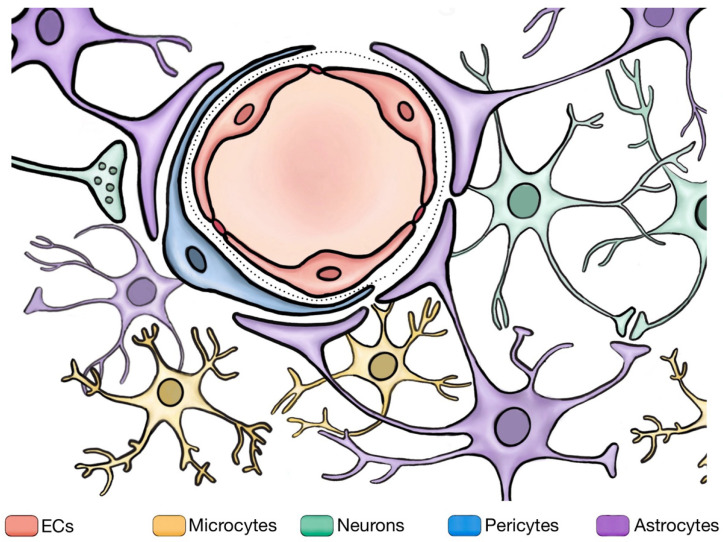
The blood–brain barrier and its neighboring cells. The figure illustrates the BBB and its interactions with neighboring cells. The BBB comprises three distinct sites. At its core, ECs form the innermost lining of cerebral blood vessels, serving as the physical barrier of the BBB. Outside these cells, pericytes play a crucial role in maintaining the BBB integrity. Surrounding the ECs and pericytes are astrocytes, which envelop them with their characteristic endfeet, providing structural and metabolic support, and modulating BBB functions by adjusting and maintaining the balance of ions, amino acids, neurotransmitters, and water in the brain. Microglia can influence the BBB permeability as the resident immune cells of the CNS. Furthermore, neurons establish synaptic connections with astrocytes, indirectly regulating the BBB integrity through signal transmission.

**Figure 3 cells-13-00260-f003:**
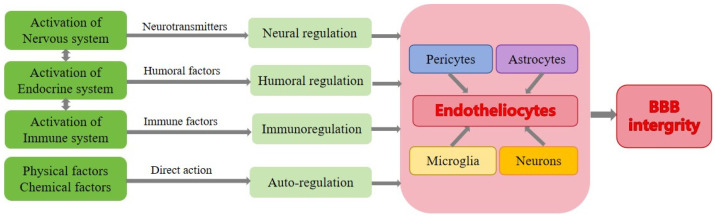
The regulation mechanisms of the BBB. The regulation of the blood–brain barrier involves several mechanisms: neuroregulation, humoral regulation, immunoregulation, and auto-regulation. These mechanisms operate at both the cellular and molecular levels, focusing on the BBB and its neighboring cells.

**Table 1 cells-13-00260-t001:** Summary of BBB damage and underlying mechanisms.

Types of Damage	Potential Causes	Main Mechanisms	Related Diseases	Representative References
Mechanical damage	Falls, transportation accidents	Disruption of tight junctions; activation of immune cells; promotion of mast cell degranulation.	Traumatic brain injury Stroke Brain tumors Epilepsy	Cash, A. and Theus, M. H. (2020) [99]. Price, L.; Wilson, C.; Grant, G. (2016) [100]. Harvey, L. A. and Close, J. C. T. (2012) [101].
Inflammatory and immune-mediated damage	Injuries, infections, neurodegenerative processes	Disruption of tight junctions; activation of immune cells; promotion of mast cell degranulation.	Multiple sclerosis Meningitis Encephalitis HIV-associated cerebral vasculitis	Skaper, S. D. (2017) [102]. Huang, X., Hussain, B., and Chang, J. (2020) [103].
Infection-related damage	Neutrophil microbes such as Neisseria meningitides viruses or bacteria-mediated damage	Release of cytokines and chemokines by immune cells; disruption of tight junctions; physical disruption; formation of Neutrophil Extracellular Traps.	Meningitis Encephalitis Neurosyphilis Toxoplasmosis AIDS	Goverdhan, P. and Pachter, J. S. (2012) [104]. H. S. L. M. (1999) [105]. Varatharaj, A. and Galea, I. (2016) [106].
Ischemic and vascular BBB damage	BBB disruption via ischemia or elevated blood pressure	BBB breakdown due to ischemia and high blood pressure, affecting ECs and basal lamina.	Ischemic stroke Hypertension	Nian, K., Harding, I. C., Herman, I. M. and Ebong, E. E. (2020) [107]. Xiaoming Hu, T., Michael De Silva, Jun Chen (2017) [108].
Toxic-factor-induced damage	Heavy metals, pesticides, chemicals, radiation	Induction of cellular damage, oxidative stress, or inflammation within BBB cells.	Diabetes mellitus Brain tumors Amyotrophic lateral sclerosis	Gundert-Remy, U. and Stahlmann, R. (2010) [109]. Jin-Tao Liu, Mo-Han Dong, Jie-Qiong Zhang (2015) [110].
Protein aggregate-related damage	Aging, genetic-related protein aggregates	Accumulation of protein aggregates in certain neurodegenerative diseases, which can directly interact with the BBB and disrupt its functions.	Alzheimer’s disease Multiple sclerosis Parkinson’s disease Huntington’s disease Prion diseases	Wu, Y.-C., Sonninen, T.-M., Peltonen, S., Koistinaho, J., and Lehtonen, Š. (2021) [111].
Mental-health induced-damage	Social isolation stress	Decreased expression of Claudin-5; microglial activation in the amygdala in female mice.	Multiple sclerosis PTSD Epilepsy	Welcome, M.O. and Mastorakis, N.E. (2020) [112]. Wu, X., Ding, Z., Fan, T. (2022) [113].

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
