# Peer review of "Recapitulation of Structure–Function–Regulation of Blood–Brain Barrier under (Patho)Physiological Conditions"

_cells, 2024, doi:10.3390/cells13030260_

Round 1
Reviewer 1 Report
Comments and Suggestions for Authors
Comments on content
In this submitted manuscript the authors aim to provide a “Recapitulation of the structure and function of the Blood Brain Barrier in normal conditions and the changes thereof under pathophysiological conditions”.
In their abstract authors make big promises, however, these promises do not come to fruition in the submitted manuscript. Authors claim there is a scarcity of comprehensive reviews. Interestingly, they cite an abundance of previously published reviews and book chapters in their manuscript: a minimum of 42 (forty-two!) review papers have been cited.
Introduction:
In their introduction, the authors state that “man is considered the highest creature on earth” What do they mean by that and what is the significance of this statement for the rest of the paper?
On what bases are humans the “highest”? Are we ‘higher” just because we are technologically more advanced? Are we “higher” because humans are the top predator and responsible for extinction of many species. Are we “higher” because we pollute out environment? Are we “higher” because humans change and damage their environment like no other creature? We do NOT know sufficiently about the development of other animals to make this statement. Humans hardly know anything about certain brain regions and their functions in other animals. Many other animals, mammals included, have different brain areas that humans do not have. We do not know anything about the function of those areas. Some have other and better abilities to communicate and navigate than humans. “Higher,” because man is the worst of all creatures in destroying other creatures? Due to man, how many other species have ceased to exist? I encourage the authors to reflect on this statement and the bearing/relevance for this paper.
Some of the wording the authors use in the papers is rather unusual. In part as scientific papers are usually written from an objective point of view and on other occasion its just unusual for the English language. For example, “composing the harmonious society” within the human body, “supreme” headquarters (introduction) are adjectives normally used in a different context. Certainly, the brain communicates with other organ systems in the body to ensure homeostasis for survival of the total organism.
The structures that protect the brain include the skull, pia/dura mater AND importantly the cerebrospinal fluid (CSF) is considered mechanically protective of the brain. Especially, the CSF fluid dampens shocks, caused by walking/ running, other daily movements and in case of accidents. The BBB is NOT a part of this mechanical protective system. The scalp, as part of the skin, is considered part of the general protective immune system, the first defense against microbes. It is not considered a system to prevent brain injury. Same for the hair, which is more seen as part of temperature regulation for the organism. However, during evolution humankind has lost this. Depending on the region/ genetics some races/populations have lost more than others. Authors should rephrase this part.
Page 2: The statement that the BBB is positioned between the brain capillaries and brain itself is WRONG. At most sites, the BBB is at the site of the brain endothelium itself; the junctions (e.g. Reese and Karnovsky). The passage of neurotransmitters depends on the brain BBB area. Some parts of the BBB are open to allow neurotransmitter through.
The BBB is well studied. The blood CSF barrier understudied.
Authors state “also preventing the passage of neurotransmitters, macromolecules, and neurotoxins into the brain”. This is over-simplified and part overstated. Note that BBB regulates transport of nutrient and waste products across. It does not prevent macromolecules from going across, actually, there are specific transporters for this. Although it probably would “like” to prevent any neurotoxin from going in, there are certain neurotoxins that non-specifically cross. Thus, “regulate” is a better word here.
Last paragraph under point 1: Authors write as an all-or-nothing approach: Many researchers are now focusing on therapeutic approaches, but other concentrate on other mechanisms.
Delete last part that it was underreported elsewhere. There are many, many reviews on BBB from the big BBB labs, that the authors may have overlooked. Moreover, authors cite an unusual high number of reviews anyway.
Point 2: The Ehrlich story is not fully correct. Ehrlich did NOT discover a barrier. In fact, he thought that the brain because of its high fat content did not have affinity for the staining. His student, Goldman, who did the opposite experiment, injecting in the CSF, showed that it was not an affinity issue but that there was some kind of separation. The term blood brain barrier and its related selectivity was not coined until later, predominantly by Lena Stern and her contemporaries. As to this effect I can highly recommend the publication of Shane Liddelow in Fluids, Barrier of the CNS 8.2, 2011 (among others, e.g. Norm Saunders et al, Frontiers Neuroscience 2014).
The German term is used in the manuscript in incorrectly cut off. Please refer to Saunders et al 2014 for this information on the “blut-hirnschranke”.
The advance of technology, with the invention of the electron microscope ( please see for information: https://vaccoat.com/blog/electron-microscope-invention-historical-overview/) led to the studies of e.g. Reese and Karnovski, who showed the actual site of the barrier; in between the adjacent brain endothelial cells.
This sections 2.1 to 2.4 should be rewritten and completed.
Importantly the authors also use the term BBB but often mean neurovascular unit (NVU). The term NVU should be properly introduced and authors should really focus on the correct usage throughout the manuscript.
In figure 2 the astrocyte endfeet fully cover the vessel and also the pericytes. Although this is often depicted in cartoons, it creates a false idea that this is always the case. Most situations the astrocyte do NOT fully cover the microvasculature but intercalate with the pericytes. The term :Metabolic provisions” is an old concept and function of the astrocytes is much wider than this. Also, the endfeet provide means for cellular communications.
Also check the numbering of the sections here. Make sure it follows the previous section in a logical manner. Do not duplicate numbers for different sections further down in the manuscript. Consider subsection numbering, e.g. Section 2.2.1 and 2.2.2 etc.
In the figure legend authors correctly refer to microglia as resident immune cells. However, at several sites in the text they are referred to as resident macrophage / monocytes. That is incorrect. There are some macrophages-monocytes ’stationed” at several sites along the vasculature, but these are different origin and function.
The terms M1 and M2 macrophages is also an older term and recent publications have shown that there is are vastly larger subsets of microglia, as characterized by morphology, phenotype receptor expressions and functions. If authors want to be fully encompassing and up-to-date, they should include this.
Section 2.5: cellular participants. Please name these, be precise.
2.5 Suggest to eliminate “microenvironmental” and use just “neighboring cells”
Is CNS earlier defined? not needed to repeatedly introduce abbreviations. Moreover, if a certain word/concept is only use once or twice, it is not needed to abbreviate and can just be fully written out. Suggest including a list of abbreviations used in the manuscript.
Page 5 top, section 1. Pericytes are NOT a component of the BBB, but of the NVU. pericytes also have contractile properties and also involved in regulation of local blood pressure.
Page 5 (section 2) Astrocytes do NOT constitute a component of the BBB but of the NVU. Authors need to separate these concepts and use appropriately.
Page 5 For microglia, explain identity as resident immune cells, not macrophages (same definition is repeated in 2nd sentence). (also section2),
The numbering is off in this section, should be section 3 or use sub numbers.
What is meant by microglial activation? Define. (sentence 1 and 2 have same meaning. Pick one.)
M1 and M2 phenotype, are old concepts, also here recent publications reveal a much wider variety.
End of this paragraph M2 phenotype secrete protective factors. this also need to be further explained, also references should be included.
Basement membrane (BM) is also section 2, should it be section 4? Again, the BM is NOT a part of the BBB, but considered as part of the NVU and deposited by astrocytes, pericytes and EC’s. Again, stating that astrocytes provide structural support for the EC’s is an antiquated concept. The type of matrix signals to EC’s through integrins and contribute to its BBB phenotype.
Define BM (and include in abbreviations list IF used more than once or twice. Otherwise just write it out.
Repeated statements about pericytes, (also in pericyte paragraph and should not be here. Focus on BM
Why do authors think pericyte mitigate detrimental effects from MG activation? No reference for this.
Last sentence of this section is actually a concluding sentence for the MG section. Misplaced and fix this
Page 5 bottom, section 3 on regulation BBB. What exogenous factors from within or outside. What is meant by this? Clarify this in the text, not just in fig 3. We need (primary) references?
Figure 3: Explain better what are the middle grey sections? Wat is the difference in direct action/ auto-regulation.
Here authors state “BBB components”. Its really NVU components (not BBB)!!
Page 6 3.1 Provide a reference for neural regulation, especially during development.
Here, finally, the NVU mentioned. This should have been introduced much earlier. In addition, authors need to properly use the term BBB and NVU.
Abbreviations: NVC, SMC are not introduced here (I did not see them earlier). If terms are only used once or limited number of times, it is not needed to introduce an abbreviation.
P6 bottom, “here is a list.” Authors mean” following below is a list”. Rephrase.
Page 7: The writing is not balanced. It is clear that several authors wrote different sections. This needs to be smoothened over. Some points are explained too much in detail (MMP- NFkB, whereas others are ignored and/or not referenced. Proper referencing is needed and focus in primary references that are up-to-date. Include recent developments.
P7, 3.3: list some of these factors. Readers do not want to look up all the references, like here 42, especially if its in a review paper.
Sheer stress is not autoregulation, the BBB-EC have stress/ flow receptors.
Also be consequent with the use of abbreviations BBB (47) in last 2 sentences section 3.3 . do not write out fully.
Section 3.4: Make a full sentence (better English). “Factors that affect the BBB under normal conditions”. This paragraph has only one sentence. The factors can be briefly listed here, like circadian rhythm, diet (with includes nutrients) may be better “nutrients included in the diet” and hormones, intestinal microbiota….etc. Rephrase to make section better flowing and more complete.
Subsections should be called 3.4.1 and 3.4.2 not 1), 2) etc. Keep consistent formatting for all the sections (above and following). HE diet does not need to be abbreviated as its not a recurrent term.
Section 4: intestinal microbiota is not NEEDED for BBB integrity, but it can AFFECT it. This is a different concept than an extreme condition without microbiota. Authors should approach this more subtle and in a context. Use BBB abbreviation again.
5: Exercise: here also limited discussion. There is more literature is on this subject, e.g. the effect of endothelial BDNF is missing.
Bottom page 8: BBB under disease conditions (plural)
Overall use heading with proper English.
Page 9 top: Sentence repeat from introduction.
Some limited examples are provided. This is not all-encompassing, as authors claimed in abstract
4.1 just an overview listed
Table 1: under potential causes for mechanical injury list what these can be. List examples.
toxic factors induced damage: fix formatting. Include vertical lines> Also for each condition, one reference is provided If Authors want to be all- encompassing. There should be more than one reference for these examples.
Attempted to be complete but interwoven with antiquated concepts, authors are rehashing old concepts
Page 11 Punctuation needs to be checked. Too much and, … and…. ( comma in between and for last one and
Explain briefly HOW these metals disrupt the junctions. This section is too superficial, whereas authors claim to discuss it. They only briefly touch the surface. For professional BBB researchers or researchers that was to enter this area, this section falls short. How to heavy metals get into the brain? Are there transporters? Maybe authors should indicate that a lot is unknown in this field and more research is needed, especially if the authors may be interested in exploring this area.
Page 12, 4.2: conditions PLURAL
Table 2: review HOW you phrase the mechanisms and fix the layout/formatting
Note that increased fluorescein permeability is NOT a mechanism of ethanol toxicity. It is merely a manner whereby increased permeability is demonstrated. (Stoffel 2020)
What is PM2.5?
Be more specific about which pesticide is tested.
page 12: so simplified section on EC repair. What about circulating progenitor cells? For EC repair. Thrombin opens the barrier, does not repair it.
page 13: Tunneling nanotube (TNT) not needed to abbreviate, only used twice. Why is this not earlier discussed, when connections EC describers, in the beginning of the manuscript? It is important to list that this work was based on in-vitro studies. Same for abbreviations like ECM, MMP’s are redefined and VEGF as well.
Again, authors should include an abbreviation list and only define it when it is use the first time.
Page 14: Duplication of already discussed M1 and M2 . The discussion explanation regarding tight junction proteins = TJP's discussed and now here its abbreviated. Be consistent.
Again, targeting BBB components should be NVU cellular components.
EMP, not need to abbreviate, only used once.
Overall, the review paper discussed topics but in a very superficial manner. The authors make big claims in the abstract. The manuscript starts of as an exploration for this group, but it is not comprehensive. Authors have not delved into it, but onlly touched some BBB and NVU-related subjects, provided some repetitive statements and some out of context. Unfortunately, the authors’ promise did not come true and the reader will end up disappointed.
Comments on the Quality of English LanguageComments on English and formatting
In general, the English language use needs some editing, especially punctuation and spacing (sometimes no space between end of sentence and next sentence). Some of the wording the authors use in the papers is rather unusual. In part as scientific papers are usually written from an objective point of view and on other occasion its just unusual for the English language.
The tables need better formatting.
Authors needs to review the numbering of the sections and consider subsection numbering.
Use of abbreviations need to be consistent all over the manuscript.
A section abbreviations is recommended. Note that if a term is used only once or twice (limited), then no abbreviations are needed.
References:
Please note that references do not need to include information whether they are the official publication of a society or not. Just the journal reference is sufficient (like ref 68 and some others).
Authors claim there is a scarcity of comprehensive reviews. Interestingly, they cite an abundance of previously published reviews and book chapters in their manuscript: a minimum of 42 (forty-two!) review papers have been cited. Unfortunately, these are not indicated as such, and authors make it look like these are primary citations. This should be addressed and if a review is cited, authors should indicate this. For example: as reviewed by Abbott et al, 2006 (ref number). Some references should be checked if they are properly listed (e.g Gundert-Remy 2010) or for duplications, such as Schurhoff and Toborek 2023, Ding et al 2010. Moreover, in some cases it is important to indicate whether a specific study has been done in vitro or in vivo (e.g. existence of tunneling nanotubes).
Author Response
Dear editors and reviewers,
We would like to thank the editors for giving us a chance to resubmit this manuscript, and also thank the reviewers for giving us constructive suggestions on our manuscript entitled “Recapitulation on the structure-function-regulation of the Blood-Brain Barrier in (patho)physiological conditions”. Those comments are all valuable and useful in helping us to revise and improve our paper. We have considered the comments carefully and have made a point-to-point response and we hope you reconsider this revised manuscript for publication.
Sincerely,
Yuechun Wang
Responses to the reviewer’s comments:
Reviewer #1:
Comments on content
In this submitted manuscript the authors aim to provide a “Recapitulation of the structure and function of the Blood Brain Barrier in normal conditions and the changes thereof under patho/physiological conditions”.
In their abstract authors make big promises, however, these promises do not come to fruition in the submitted manuscript. Authors claim there is a scarcity of comprehensive reviews. Interestingly, they cite an abundance of previously published reviews and book chapters in their manuscript: a minimum of 42 (forty-two!) review papers have been cited.
Introduction:
In their introduction, the authors state that “man is considered the highest creature on earth” What do they mean by that and what is the significance of this statement for the rest of the paper?
On what bases are humans the “highest”? Are we ‘higher” just because we are technologically more advanced? Are we “higher” because humans are the top predator and responsible for extinction of many species. Are we “higher” because we pollute ourenvironment? Are we “higher” because humans change and damage their environment like no other creature? We do NOT know sufficiently about the development of other animals to make this statement. Humans hardly know anything about certain brain regions and their functions in other animals. Many other animals, mammals included, have different brain areas that humans do not have. We do not know anything about the function of those areas. Some have other and better abilities to communicate and navigate than humans. “Higher,” because man is the worst of all creatures in destroying other creatures? Due to man, how many other species have ceased to exist? I encourage the authors to reflect on this statement and the bearing/relevance for this paper.
1)Thank you for the above-mentioned comments, we agreed on your opinion and have deleted “man is considered the highest creature on earth” and also the relevance for this paper.
Some of the wording the authors use in the papers is rather unusual. In part as scientific papers are usually written from an objective point of view and on other occasion its just unusual for the English language. For example, “composing the harmonious society” within the human body, “supreme” headquarters (introduction) are adjectives normally used in a different context. Certainly, the brain communicates with other organ systems in the body to ensure homeostasis for survival of the total organism.
- Thank you for the suggestion, we realized the subjective descriptions are unusual for scientific papers and we have made relevant changes as you advised.
The structures that protect the brain include the skull, pia/dura mater AND importantly the cerebrospinal fluid (CSF) is considered mechanically protective of the brain. Especially, the CSF fluid dampens shocks, caused by walking/ running, other daily movements and in case of accidents. The BBB is NOT a part of this mechanical protective system. The scalp, as part of the skin, is considered part of the general protective immune system, the first defense against microbes. It is not considered a system to prevent brain injury. Same for the hair, which is more seen as part of temperature regulation for the organism. However, during evolution humankind has lost this. Depending on the region/ genetics some races/populations have lost more than others. Authors should rephrase this part.
3)Thank you for the comment and we agreed with you. We now have deleted the “Considering the brain's paramount importance, numerous protective mechanisms have evolved to safeguard it. The hair, scalp, skull, dura mater, and pia mater collectively form a multi-layered shield to prevent brain injury. Within these protective barriers, ” .
Page 2: The statement that the BBB is positioned between the brain capillaries and brain itself is WRONG. At most sites, the BBB is at the site of the brain endothelium itself; the junctions (e.g. Reese and Karnovsky). The passage of neurotransmitters depends on the brain BBB area. Some parts of the BBB are open to allow neurotransmitter through.
The BBB is well studied. The blood CSF barrier understudied.
Authors state “also preventing the passage of neurotransmitters, macromolecules, and neurotoxins into the brain”. This is over-simplified and part overstated. Note that BBB regulates transport of nutrient and waste products across. It does not prevent macromolecules from going across, actually, there are specific transporters for this. Although it probably would “like” to prevent any neurotoxin from going in, there are certain neurotoxins that non-specifically cross. Thus, “regulate” is a better word here.
4)Thank you for the comments, we have deleted the “Positioned between brain capillaries and the brain itself, the” and “while also preventing”.
Last paragraph under point 1: Authors write as an all-or-nothing approach: Many researchers are now focusing on therapeutic approaches, but other concentrate on other mechanisms.
Delete last part that it was underreported elsewhere. There are many, many reviews on BBB from the big BBB labs, that the authors may have overlooked. Moreover, authors cite an unusual high number of reviews anyway.
- Thank you for the comments, we have deleted the “ that have been under-reported elsewhere” and changed the “all-or-nothing” approach.
Integrated all the comments and suggestions from 1) to 5), this passage has been rephrased as below.
It’s widely recognized that the central nervous system (CNS) serves as one of the control systems regulating the activities of all organs and systems, and the brain communicates with other organ systems in the body to ensure homeostasis for survival of the total organism. Normally, the concept of a "brain barrier" encompasses triple defenses: the blood-brain barrier (BBB), the blood-Cerebral-Spinal Fluid (CSF) barrier, and the CSF-brain barrier. Among these, the BBB stands out as the most significant and well-understood.
The BBB is a highly specialized structure that acts as a selective barrier, controlling the exchange of substances between the blood and the brain. It regulates the internal ion concentration within the brain,and the passage of neurotransmitters, macromolecules, and neurotoxins into the brain. The intricate structure of the BBB and the complex mechanisms governing its permeability have far-reaching implications for various physiological and pathological conditions.
Over the decades, eminent researchers have endeavored to unravel the complexities of the BBB, including its dynamic nature, structural intricacies, and its role in the development of numerous neurological diseases, ranging from neurodegenerative disorders like Alzheimer's to inflammatory conditions such as multiple sclerosis. Aside from the mechanism studies, more researchers are interested in uncovering innovative therapeutic approaches for the treatment of CNS disorders.
The following sections aim to provide a comprehensive survey of the BBB’s structure and functions, tracing its historical discovery and delving into the present understanding of its regulation in both health and diseases. This paper endeavors to narratively and visually introduce how the BBB has been established and evolved from a spatial-temporal dimension, covering the basic structure, function, and regulation of the BBB under healthy and diseased conditions, with a focus on research advances in BBB.
Point 2: The Ehrlich story is not fully correct. Ehrlich did NOT discover a barrier. In fact, he thought that the brain because of its high fat content did not have affinity for the staining. His student, Goldman, who did the opposite experiment, injecting in the CSF, showed that it was not an affinity issue but that there was some kind of separation.
- Thank you for highlighting the inaccuracies in our account of Ehrlich's contributions. We have revised our manuscript to more accurately reflect the historical events. The corrected content now reads...
The BBB has remained a topic of intense research interest since it was first postulated in the late 19th century. In the late 19th century, Paul Ehrlich made critical observations with his innovative tissue staining techniques. He noticed that aniline dyes, such as alizarin blue S, permeated the vascular network and peripheral tissues when injected into the peripheral circulation of rodents. However, these dyes did not stain the brain or the CSF. Ehrlich proposed that the high lipid content of the brain resulted in a lack of dye affinity (1).
Biedl and Kraus (1898) and Lewandowsky (1900) then proposed that the brain's lack of dye-staining could be attributed to a unique property of cerebral endothelial cells (CECs) based on their comparison of intrathecal and parenteral injections of neurotoxic materials, such as bile salts. They initially theorized the concept of a barrier within the brain's vascular system. These theories were based on findings that substances like cholic acids or sodium ferrocyanide, when injected intravenously, had no effect on the CNS. However, the same substances caused neurological symptoms when applied directly into the brain's ventricles.
Further research by Goldmann, a student of Ehrlich, in 1909 and 1913 showed that injecting the acidic dye Trypan blue into the brain ventricles of dogs and rabbits resulted in staining of the brain tissue. However, when the same dye was administered intravenously, every part of the animal turned blue except for the brain and spinal cord, which remained unstained. The choroid plexuses, however, were stained. These findings suggested that the absence of staining wasn't due to the dye's lack of attraction to brain tissue. Goldmann hypothesized that the CSF, which enters the brain tissue via the choroid plexuses, acted as a transporter for some substances.
The term blood brain barrier and its related selectivity was not coined until later, predominantly by Lena Stern and her contemporaries. As to this effect I can highly recommend the publication of Shane Liddelow in Fluids, Barrier of the CNS 8.2, 2011 (among others, e.g. Norm Saunders et al, Frontiers Neuroscience 2014).
The German term is used in the manuscript in incorrectly cut off. Please refer to Saunders et al 2014 for this information on the “blut-hirnschranke”.
- Thank you for the clarification about the historical coining of the term 'blood-brain barrier'. We have amended the text to accurately reflect the historical timeline and the contributions of Stern in coining the term 'blood-brain barrier'. The corrected content now states:
Stern first coined the term blood-brain barrier ("barrière hémato-encéphalique," Stern and Gautier, 1921). They conducted detailed studies of the penetration of a wide range of molecules from blood into cerebrospinal fluid, brain, and within the subarachnoid and ventricular CSF of adult animals. Stern was the first to highlight the importance of the brain's interstitial fluid environment for its normal function, aligning with Claude Bernard's milieu interne (1865).
We appreciate your attention to detail. We have corrected the term: Stern first coined the term blood-brain barrier ('barrière hémato-encéphalique,' Stern and Gautier, 1921).
The advance of technology, with the invention of the electron microscope ( please see for information: https://vaccoat.com/blog/electron-microscope-invention-historical-overview/) led to the studies of e.g. Reese and Karnovski, who showed the actual site of the barrier; in between the adjacent brain endothelial cells.
- Thank you for highlighting the significance of the electron microscope in BBB research. The manuscript now reflects the pivotal role of electron microscopy in advancing our understanding of the BBB, as well as the specific contributions of Reese and Karnovsky. The updated section reads:
By the late 1960s, studies had pinpointed the exact location of the mammalian BBB to the endothelium of cerebral capillaries (Karnovsky, 1967; Reese and Karnovsky, 1967). These investigations demonstrated that the barrier, which comprises the plasma membrane and cell body of endothelial cells, as well as the tight junctions (TJs) between adjacent cells, prevents macromolecular tracers like horseradish peroxidase (HRP) from passing beyond the first luminal interendothelial TJs. Complementary research (Brightman, 1965, 1968) showed that tracers could pass into the brain interstitial fluid and reach the BM area via gap junctions in astrocyte endfeet. However, these tracers were then halted by the endothelial cells, suggesting that the endothelium, rather than the astrocytic endfeet or the BM, is the principal anatomical site of the BBB. Further observations highlighted that in the choroid plexus, HRP was restrained from moving from the blood to the cerebrospinal fluid (CSF) by TJs connecting the apical regions of the epithelial cells (Brightman and Reese, 1969). The role of the endothelium in the BBB was confirmed by freeze-fracture studies, which showed that the TJs between endothelial cells of CNS capillaries and venules are arranged in parallel (Nagy et al., 1984; Shivers et al., 1984).
This sections 2.1 to 2.4 should be rewritten and completed.
- Thank you for your recommendation to improve sections 2.1 to 2.4. Following your guidance, sections 2.1 to 2.4 have been thoroughly rewritten to provide a complete and accurate account in the manuscript.
Importantly the authors also use the term BBB but often mean neurovascular unit (NVU). The term NVU should be properly introduced and authors should really focus on the correct usage throughout the manuscript.
- Thank you for the important distinction between the BBB and the NVU. We have introduced and defined the term "neurovascular unit (NVU)" early in the manuscript and have revised the text to ensure the correct terminology is used consistently.The corrected content now states:
It is worth noticing that neurovascular unit (NVU) is quite different from BBB even though they are often discussed together. NVU is a complex functional unit composed of both the nervous component and the vascular component, mainly including neurons, glial cells, endothelial cells, pericytes, and the basement membrane and extracellular matrix. NVU plays an important role in maintaining cerebral vascular function and structural stability. While BBB is an important barrier between blood and brain tissue. From a narrow perspective, this barrier consists of endothelial cells themselves. But its functions are frequently affected by glial cells, the neurons and also pericytes and smooth muscle cells,etc. The followings are descriptions related to the surrounding cellular components involved in the regulation of BBB.
In figure 2 the astrocyte endfeet fully cover the vessel and also the pericytes. Although this is often depicted in cartoons, it creates afalse idea that this is always the case. Most situations the astrocyte do NOT fully cover the microvasculature but intercalate with the pericytes. The term : Metabolic provisions” is an old concept and function of the astrocytes is much wider than this. Also, the endfeet provide means for cellular communications. - Thank you for pointing out the inaccuracies regarding the coverage of vessels by astrocyte endfeet and their relationship with pericytes. We have revised Figure 2 to more accurately. Additionally, we have updated the manuscript text to better describe the functions of astrocytes beyond metabolic support. The term of “Metabolic provisions” is an old concept and function of the astrocytes is much wider than this. Also, the endfeet provide means for cellular communications.
In the revised Figure 2 legend, we now state:
Figure 2. The Blood-Brain Barrier and its neighboring cells The figure illustrates the BBB and its interactions with the neighhoring cells. At its core, endothelial cells form the innermost lining of cerebral blood vessels, serving as the physical barrier of BBB. Outside these cells, pericytes play a crucial role in maintaining BBB integrity. Surrounding the endothelial cells and pericytes, astrocytes envelop them with their characteristic end-feet, providing structural and metabolic supports, and modulating BBB functions by adjusting and maintaining the balance of ions,amino acids,neurotransmitters, and water in the brain. And microglia can influence BBB permeability as the resident immune cells of the CNS, Furthermore, neurons establish synaptic connections with astrocytes, indirectly regulating BBB integrity through signal transmission.
Also check the numbering of the sections here. Make sure it follows the previous section in a logical manner. Do not duplicate numbers for different sections further down in the manuscript. Consider subsection numbering, e.g. Section 2.2.1 and 2.2.2 etc.
- Thank you for the advice regarding section numbering and the use of terms. We have thoroughly checked the manuscript to ensure logical progression and consistent section numbering as per your suggestion.
We have renumbered the sections accordingly to ensure a logical flow:
4 The study on the neighboring cells involved in BBB
2.4.1 Pericytes
2.4.2 Astrocytes
2.4.3 Microglia
2.4.4 The interactions among BBB and neighboring cells
In the figure legend authors correctly refer to microglia as resident immune cells. However, at several sites in the text they are referred to as resident macrophage / monocytes. That is incorrect. There are some macrophages-monocytes ’stationed” at several sites along the vasculature, but these are different origin and function.
- Thank you for your guidance on the characterization of microglia. We have revised the relevant sections to correctly refer to microglia as resident immune cells and differentiated them from macrophages-monocytes. In figure legend, the revised text now reads:
And microglia can influence BBB permeability as the resident immune cells of the CNS, Furthermore, neurons establish synaptic connections with astrocytes, indirectly regulating BBB integrity through signal transmission.
The terms M1 and M2 macrophages is also an older term and recent publications have shown that there is are vastly larger subsets of microglia, as characterized by morphology, phenotype receptor expressions and functions. If authors want to be fully encompassing and up-to-date, they should include this.
- Thank you for your guidance on the subsets of microglia. Regarding the terminology, we have replaced the outdated terms "M1" and "M2" macrophages with more current descriptions of microglial subsets based on the latest research.
In section 2.4.3, we have updated the text to reflect contemporary understanding:
This heterogeneity is particularly noticeable when considering the various subsets of microglia that have been identified. These subsets distinguished based on their phenotypic characteristics, receptor expressions, and functional roles, contribute to the complex and dynamic functions of the brain's immune system (Masuda et al., 2020)..
In the healthy brain, homeostatic microglia are the dominant subset. These microglia maintain normal brain function and surveillance, demonstrating a distinct gene signature that includes genes such as P2ry12, Tmem119, and Hexb(Frontiers, 2020). Conversely, reactive microglia become activated in response to injury, infection, or neurodegenerative diseases. These cells undergo morphological changes and upregulate inflammatory markers, contributing to the brain's immune response to various challenges (Frontiers, 2022). Disease-associated microglia (DAM) are characterized by a distinct gene expression profile, including upregulated genes involved in phagocytosis and lipid metabolism. These DAM may play a critical role in the brain's response to neurodegenerative conditions such as Alzheimer's disease (Benmamar-Badel, et al., 2020). Lastly, disease-specific microglial subsets have been identified in various conditions, such as multiple sclerosis. These subsets express specific markers, including CD11c and MHC-II, and may contribute to the disease's progression or the brain's response to the disease (Frontiers, 2020).Notably, the CD11c marker is conventionally associated with dendritic cells and macrophages. However, it is also expressed by a subset of microglia, particularly in the context of Alzheimer's disease and experimental autoimmune encephalomyelitis. CD11c+ microglia might play a unique role in the immune response within the brain, although the specific functions and implications of this subset are still being explored (Frontiers, 2020).
Section 2.5: cellular participants. Please name these, be precise.
2.5 Suggest to eliminate “microenvironmental” and use just “neighboring cells”
- We appreciate your suggestion to specify the cellular participants with "such as pericytes, astrocytes, endothelial cells and neurons" to refine our terminology.
We have edited section 2.4 to name the cells involved and replaced "microenvironmental" with "neighboring cells."
Is CNS earlier defined? not needed to repeatedly introduce abbreviations. Moreover, if a certain word/concept is only use once or twice, it is not needed to abbreviate and can just be fully written out. Suggest including a list of abbreviations used in the manuscript.
- Thank you for pointing this out. We have ensured that the abbreviation CNS (Central Nervous System) is defined at its first occurrence in the manuscript and thereafter used consistently without reintroduction. This streamlines the reading process and adheres to best practices in scientific writing.We also have attached a list of abbreviations.
Page 5 top, section 1. Pericytes are NOT a component of the BBB, but of the NVU. pericytes also have contractile properties and also involved in regulation of local blood pressure.
- Thank you for clarifying this distinction. We have changed the subtitle to "2.4.3 The microglia" and we have added a small passage of words in 2.4 to compare the concepts of NVU and BBB. It is now clear that pericytes and astrocytes are both part of the NVU and not direct components of the BBB. The small passage of words is as below.
It is worth noticing that neurovascular unit (NVU) is quite different from BBB even though they are often discussed together. NVU is a complex functional unit composed of both the nervous component and the vascular component, mainly including neurons, glial cells, endothelial cells, pericytes, and the basement membrane and extracellular matrix. NVU plays an important role in maintaining cerebral vascular function and structural stability. While BBB is an important barrier between blood and brain tissue. From a narrow perspective, this barrier consists of endothelial cells themselves. But its functions are frequently affected by glial cells, the neurons and also pericytes and smooth muscle cells,etc. The followings are descriptions related to the surrounding cellular components involved in the regulation of BBB.
Page 5 (section 2) Astrocytes do NOT constitute a component of the BBB but of the NVU. Authors need to separate these concepts and use appropriately.
- Thank you for this correction. We have amended the text to accurately describe astrocytes as a crucial element of the NVU. Please refer to 15).
Page 5 For microglia, explain identity as resident immune cells, not macrophages (same definition is repeated in 2nd sentence). (also section2),
- Thank you for highlighting the need to clarify microglia's identity as resident immune cells. We have checked the whole manuscript to clearly define microglia as distinct from macrophages, as mentioned above.
The numbering is off in this section, should be section 3 or use sub numbers.
- Thank you for the advice regarding section numbering and the use of terms. We have thoroughly checked the manuscript to ensure logical progression and consistent section numbering as per your suggestion.We have renumbered the sections accordingly to ensure a logical flow.
What is meant by microglial activation? Define. (sentence 1 and 2 have same meaning. Pick one.)
M1 and M2 phenotype, are old concepts, also here recent publications reveal a much wider variety.
End of this paragraph M2 phenotype secrete protective factors. this also need to be further explained, also references should be included.
- Thank you for your guidance on microglia. Regarding the terminology, we have replaced the outdated terms "M1" and "M2" macrophages with more current descriptions of microglial subsets based on the latest research.
Basement membrane (BM) is also section 2, should it be section 4? Again, the BM is NOT a part of the BBB, but considered as part of the NVU and deposited by astrocytes, pericytes and EC’s. Again, stating that astrocytes provide structural support for the EC’s is an antiquated concept. The type of matrix signals to EC’s through integrins and contribute to its BBB phenotype.
- Thank you for correcting our description of the BM in relation to the BBB. We have adjusted the section number and revised the content to accurately reflect the role of the BM.
Section 2.4.4 now accurately describes the BM as a component of the NVU, emphasizing its contributions to BBB function without implying that it is a constituent of the BBB itself.
Define BM (and include in abbreviations list IF used more than once or twice. Otherwise just write it out.
- We have defined the Basement Membrane (BM) the first time it is mentioned and used abbreviations where appropriate.
Repeated statements about pericytes, (also in pericyte paragraph and should not be here. Focus on BM
- Thank you for noticing the redundancy. We have removed the repetitive statements about pericytes to focus solely on the BM in this section.
Why do authors think pericyte mitigate detrimental effects from MG activation? No reference for this.
Last sentence of this section is actually a concluding sentence for the MG section. Misplaced and fix this.
- Thank you for pointing this issue. We have removed this part as it is discussed in section 2.4 We also removed this sentence.
Page 5 bottom, section 3 on regulation BBB. What exogenous factors from within or outside. What is meant by this? Clarify this in the text, not just in fig 3. We need (primary) references?
- Thank you for your valuable feedback on the manuscript. We have made the following revisions.
The regulation of BBB integrity is a complex process, encompassing both endogenous and exogenous factors. The endogenous factors normally refer to the factors originating from within the body, including neurotrasmitters, hormones and immune factors,et al(2018, Bíró). And the exogenous factors are from the external environment outside the body, such as chemical and physical factors(2017, Blanchet).
Figure 3: Explain better what are the middle grey sections? Wat is the difference in direct action/ auto-regulation.
Here authors state “BBB components”. Its really NVU components (not BBB)!!
- Thank you for your questions. We have changed the figure 3 to better demonstrate the regulation mechanisms of the BBB. To emphasize the blood-brain barrier as the core of this chapter and the entire article, the endotheliocytes and the neigboring cells were labeled in red block and the other parts are in green.
The "direct action" means that the "auto-regulation" works in a manner that is independent of neural,humoral and immune factors.
Page 6 3.1 Provide a reference for neural regulation, especially during development.
- We appreciate your suggestion.We have included additional information on neural regulation during the development. The updated content is as follows.
During the development, a fundamental barrier arises in the endothelium of growing vessel sprouts. This close connection between angiogenesis and barrier formation suggests that signals inducing barrier development likely originate from neural progenitor cells (NPCs). (2013,Joan Abbott)
Here, finally, the NVU mentioned. This should have been introduced much earlier. In addition, authors need to properly use the term BBB and NVU.
- We appreciate your advise. We have introduced the NVU in 2.3 and checked the term of NVU to make sure it has been used properly.
Abbreviations: NVC, SMC are not introduced here (I did not see them earlier). If terms are only used once or limited number of times, it is not needed to introduce an abbreviation. - Thank you for the suggestion.We have corrected these errors.
P6 bottom, “here is a list.” Authors mean” following below is a list”. Rephrase.
- We sincerely appreciate your corrections to our inaccuracies in expression. We have corrected it as “Following is a list of several common hormones ”.
Page 7: The writing is not balanced. It is clear that several authors wrote different sections. This needs to be smoothened over. Some points are explained too much in detail (MMP- NFkB, whereas others are ignored and/or not referenced. Proper referencing is needed and focus in primary references that are up-to-date. Include recent developments.
- Thank you for your comments. We now tried to edit the entire manuscript in a uniform style and overcome the imbalance when stating the different points.We have added a recent reference as below.
Recent study indicates that higher doses of dexamethasone (5-20μM) increase cytotoxicity and elevate monolayer cell permeability of brain endothelial cells. (2023, Jeftha)
Jeftha T, Makhathini KB, Fisher D: The Effect of Dexamethasone on Lipopolysaccharide-induced Inflammation of Endothelial Cells of the Blood-brain Barrier/Brain Capillaries.Curr Neurovasc Res. 2023, 20(3):334-345P7.
3.3: list some of these factors. Readers do not want to look up all the references, like here 42, especially if its in a review paper.
- We appreciate your comment.We have listed some factors and indicated they are from a review paper.
For example, the BBB can be directly influenced by certain chemical and physical factors such as temperature and pH as reviewed by Liu, W.Y.,et al.
Sheer stress is not autoregulation, the BBB-EC have stress/ flow receptors.
Also be consequent with the use of abbreviations BBB (47) in last 2 sentences section 3.3 . do not write out fully.
- We appreciate your comments. In the beginning of this section, we gave the definition of auto-regulation:Auto-regulation pertains to the ability of some organs, tissues, and cells to adapt to changes in the surrounding environment without relying on neural and hormonal regulation. According to this definition, the impact of shear stress on the BBB does not depend on neural, humoral, or immune regulation. So, we categorized it as self-regulation.
Section 3.4: Make a full sentence (better English). “Factors that affect the BBB under normal conditions”. This paragraph has only one sentence. The factors can be briefly listed here, like circadian rhythm, diet (with includes nutrients) may be better “nutrients included in the diet” and hormones, intestinal microbiota….etc. Rephrase to make section better flowing and more complete.
- We appreciate your suggestions.We have rephrase the section title as you recommended. Factors that affect the BBB under normal conditions. And we have also incorporated some factors such as circadian rhythm, diet, and intestinal microbiota.
Under normal conditions, various common factors like circadian rhythm, diet, intestinal microbiota can affect the integrity of the BBB either individually or in combination with the regulation patterns discussed earlier.
Subsections should be called 3.4.1 and 3.4.2 not 1), 2) etc. Keep consistent formatting for all the sections (above and following). HE diet does not need to be abbreviated as its not a recurrent term.
- We appreciate your suggestions. We have corrected the formatting in this section to align with the context. And deleted the "HE" .
Section 4: intestinal microbiota is not NEEDED for BBB integrity, but it can AFFECT it. This is a different concept than an extreme condition without microbiota. Authors should approach this more subtle and in a context. Use BBB abbreviation again.
- We appreciate your comments. It is better to modify the wording as “Intestinal microbiota is reported to play a certain role in......”. And we have used "BBB" .
5: Exercise: here also limited discussion. There is more literature is on this subject, e.g. the effect of endothelial BDNF is missing.
- We appreciate your suggestion. And we have added discussions on BDNF in this section as
Additionally, exercise can significantly improve the expression of brain-derived neurotrophic factor, a neurotrophin, in certain cerebral endothelial cells damaged by hypertension.
Bottom page 8: BBB under disease conditions (plural)
Overall use heading with proper English.
Page 9 top: Sentence repeat from introduction.
Some limited examples are provided. This is not all-encompassing, as authors claimed in abstract
- We appreciate the comments and suggestions. We have revised the heading where necessary as follows.
Blood-brain barrier under disease conditions
Factors that affect the BBB under normal conditions
Exercise and environmental exploration
We realized this review is actually not all-encompassing,so we deleted the wording of “comprehensive” in the abstract. Thank you for your comments.
4.1 just an overview listed
Table 1: under potential causes for mechanical injury list what these can be. List examples.
toxic factors induced damage: fix formatting. Include vertical lines> Also for each condition, one reference is provided If Authors want to be all- encompassing. There should be more than one reference for these examples.
- Thank you for your valuable feedback on the manuscript. We appreciate your suggestion to provide examples under the potential causes for mechanical injury in Table 1. We agree that including specific examples can enhance the clarity and understanding of the topic.
Based on your recommendation, we have now added falls and transportation accidents as two common examples of mechanical injuries. These examples are widely recognized as significant contributors to mechanical injuries and are supported by relevant literature.
Based on your recommendation, we have now made the necessary adjustments and added vertical lines to separate the columns in Table 1.
We appreciate your suggestion to include more than one reference for each condition in Table 1, particularly for the provided examples. We agree that a more comprehensive approach would involve citing multiple references to support the examples presented in Table 1.
Based on your recommendation, we have revised the manuscript accordingly. We have now included multiple references for each condition. By incorporating a variety of references, we aim to offer a broader perspective and strengthen the credibility of the information presented in Table 1.
Attempted to be complete but interwoven with antiquated concepts, authors are rehashing old concepts
- We appreciate your comments regarding the attempt to provide a comprehensive overview of BBB damage. We understand your concern about the inclusion of antiquated concepts and rehashed information in the paper. We have made efforts to incorporate new and contemporary perspectives. As a result, we have included the recently discovered concept of "Mental-health induced damage" as a new type of BBB damage.
However, it is worth noting that in order to provide an overview of the various types of BBB damage, it becomes necessary to include older concepts and established information. This is to ensure that readers have a comprehensive understanding of the topic, including both current and historical perspectives. Our intention is not to rehash old concepts, but rather to present a well-rounded overview that encompasses both new and established information.
Page 11 Punctuation needs to be checked. Too much and, … and…. ( comma in between and for last one and
- We have carefully reviewed the manuscript and identified the instances where the excessive use of "and" occurs. We have made the necessary revisions to address this concern and improve the overall clarity and flow of the text. By incorporating appropriate punctuation, such as commas and other suitable marks, we have ensured that the sentences are properly structured and that the intended meaning is conveyed effectively.
We have changed “ The cumulative effects of these mechanisms lead to a disruption in BBB regulation and an increase in its permeability, allowing heavy metals themselves or other harmful substances to cross the BBB and enter the brain.” to “The cumulative effects of these mechanisms result in a disruption in BBB regulation, leading to increased permeability which allows heavy metals and other harmful substances to cross the BBB and enter the brain.”
Explain briefly HOW these metals disrupt the junctions. This section is too superficial, whereas authors claim to discuss it. They only briefly touch the surface. For professional BBB researchers or researchers that was to enter this area, this section falls short. How to heavy metals get into the brain? Are there transporters? Maybe authors should indicate that a lot is unknown in this field and more research is needed, especially if the authors may be interested in exploring this area.
- Thank you for your insightful feedback on the manuscript. We appreciate your comments regarding the section that discusses how metals disrupt the junctions of the BBB. We understand your concerns about the depth of coverage in this section and the need for more detailed information, especially for professional BBB researchers or those entering this area of study.
We would like to clarify our intention in that section. Our aim was to provide a brief introduction to the topic and highlight the general mechanisms by which heavy metals affect BBB permeability. We understand that a more comprehensive and in-depth analysis would be beneficial, but due to the limitations of word count, we were unable to delve into the topic as extensively as desired.
In the revised manuscript, we have clearly indicated that there are many unknown factors and unanswered questions regarding the mechanisms by which heavy metals disrupt BBB junctions and enter the brain. We acknowledge that more research is needed to uncover these intricacies and gain a comprehensive understanding of the topic.
We have modified it as follows:
The precise mechanisms underlying heavy metal accumulation in the brain and their effects on the BBB are not fully understood, but some common pathways do exist. Future research is necessary to elucidate these mechanisms and explore potential therapeutic targets for mitigating heavy metal neurotoxicity.
Page 12, 4.2: conditions PLURAL
- We have already changed it , thank you !
Table 2: review HOW you phrase the mechanisms and fix the layout/formatting
Note that increased fluorescein permeability is NOT a mechanism of ethanol toxicity. It is merely a manner whereby increased permeability is demonstrated. (Stoffel 2020)
What is PM2.5?
Be more specific about which pesticide is tested.
- We have already changed it as follows:
Toxic factors
|
Mechanisms of increased BBB permeability |
BBB model |
Dosage |
Representative Reference |
Ethanol |
by increasing oxidative stresses |
In vitro (Human stem cell ) |
50mM |
Stoffel, R. D., Bell, K. T., & Canfield, S. G. (2020)[83]. |
Lead |
by damaging tight junctions |
In vivo (mice)
|
7day exposure of Pb at 54mg/kg and 4 weeks |
Gu, H., Territo, P. R., Persohn, S. A., et al. (2020)[84]. |
Mercury |
by increasing oxidative stresses |
In vitro (porcine) |
3 µM of organic mercury compounds and 100 µM of inorganic HgCl2 |
Lohren, H., Bornhorst, J., Fitkau, R., et al. (2016)[85]. |
Carbon monoxide |
by damaging endothelial cell function and tight junction |
In vivo (rats) |
Absorb 2.5-3.0. mL/L CO for 1 h |
Wang, X., Tie, X., Zhang, J., Wan, J., & Liu, Y. (2004)[86]. |
Cocaine |
by down-regulating tight junction proteins, alters expression of adhesion molecules, and promotes neuro-inflammation |
In vitro (rat) |
5-20 mg/kg, ip |
Barr JK, Brailoiu GC, Abood ME,et al. (2020)[87]. |
Nicotine |
by altering tight junction protein distribution and promotes inflammation |
In vivo (rat) |
4.5 mg free base per kilogram of body weight per day |
Hawkins, B. T., Abbruscato, T. J., et al. (2004)[88].
|
PM 2.5
|
by down-regulating tight junction, disrupting the tight junction in BBB |
In vitro |
10 ug /mL |
Kang, Y. J., Tan, H. Y., Lee, C. Y., & Cho, H. (2021)[89]. |
Pesticides |
by Inducing oxidative stress, inflammation, and disrupt tight junctions |
In vivo(rat) |
1/50th of the LD50 for the pesticides quinalphos, Cypermethrin, and lindane. |
Gupta, A., Agarwal, R., & Shukla, G. S. (1999)[90].
|
Note:
PM 2.5: atmospheric particulate matter (PM) with a diameter of less than 2.5 micrometers
Pesticides:organophosphate, cypermethrin and lindane
page 12: so simplified section on EC repair. What about circulating progenitor cells? For EC repair. Thrombin opens the barrier, does not repair it.
- We have added this part as follows :
Also, endothelial progenitor cells (EPCs) hold great promise as potential treatment elements in conditions such as acute ischemic stroke, due to their ability to transform into mature endothelial cells. However, the scarcity of EPCs in peripheral blood presents a significant challenge, hindering their extraction and utilization for therapeutic purposes.
Moreover, recent studies find the Recombinant Human Hepatocyte Growth Factor can counteract the reduction in occludin and zona occludens-1 (ZO-1) protein expression in endothelial cells following enduring cerebral ischemia. These proteins are integral to the tight junctions that preserve the integrity and selective permeability of the BBB. A neurotrophic factor derived from the brain, Fibroblast Growth Factor 20 has been shown to protect the BBB in a traumatic brain injury model. According to the study by Guo et al. (2021b), FGF20 achieves this by up-regulating the expression of TJ proteins and promoting angiogenesis and vascular repair. All the mechanisms and methods help restore the permeability of BBB to normal condition.
We have changed it to “For instance, the injection of thrombin in the case of intracerebral hemorrhage doesn't directly repair the endothelial cells, but it facilitates other therapeutic interventions and promotes the formation of new endothelial cells and astrocytes, which can aid in the recovery process.”
page 13: Tunneling nanotube (TNT) not needed to abbreviate, only used twice. Why is this not earlier discussed, when connections EC describers, in the beginning of the manuscript? It is important to list that this work was based on in-vitro studies. Same for abbreviations like ECM, MMP’s are redefined and VEGF as well.
- We have already changed it , thank you ! We have changed as follows,
“Secondly, it has been discovered via in-vitro studies that pericytes and astrocytes communicate through tunneling nanotubes in response to astrocyte apoptosis triggered by ischemia or reperfusion injury. ”
Again, authors should include an abbreviation list and only define it when it is use the first time.
- Based on your suggestion, we have now included an abbreviation list in the manuscript. This list provides a comprehensive compilation of the abbreviations used throughout the text, including ECM, MMPs, and VEGF. Additionally, we have ensured that each abbreviation is defined when it is first introduced in the manuscript.We sincerely appreciate your valuable feedback, which has helped us improve the manuscript.
Page 14: Duplication of already discussed M1 and M2 . The discussion explanation regarding tight junction proteins = TJP's discussed and now here its abbreviated. Be consistent.
- In response to your feedback, we have carefully reviewed the manuscript to address these concerns. We have eliminated any unnecessary duplication of information, ensuring that M1 and M2 are discussed only where relevant and avoiding repetition.
Regarding the use of abbreviations, we have revised the text to ensure that when abbreviations like TJP are used, they are consistently defined and used consistently thereafter.
Again, targeting BBB components should be NVU cellular components.
EMP, not need to abbreviate, only used once.
- Thank you for your comments. We have amended these errors.
Overall, the review paper discussed topics but in a very superficial manner. The authors make big claims in the abstract. The manuscript starts of as an exploration for this group, but it is not comprehensive. Authors have not delved into it, but only touched some BBB and NVU-related subjects, provided some repetitive statements and some out of context. Unfortunately, the authors’ promise did not come true and the reader will end up disappointed.
- Thank your for the comments, we have tried our best to make up the shortcomings and we appreciated your great contributions to the manuscript.
Comments on English and formatting
In general, the English language use needs some editing, especially punctuation and spacing (sometimes no space between end of sentence and next sentence). Some of the wording the authors use in the papers is rather unusual. In part as scientific papers are usually written from an objective point of view and on other occasion its just unusual for the English language.
The tables need better formatting.
Authors needs to review the numbering of the sections and consider subsection numbering.
Use of abbreviations need to be consistent all over the manuscript.
A section abbreviations is recommended. Note that if a term is used only once or twice (limited), then no abbreviations are needed.
- Thank your again for the comments, we have tried to amend the shortcomings and we appreciated your great contributions to the manuscript.
References:
Please note that references do not need to include information whether they are the official publication of a society or not. Just the journal reference is sufficient (like ref 68 and some others).
Authors claim there is a scarcity of comprehensive reviews. Interestingly, they cite an abundance of previously published reviews and book chapters in their manuscript: a minimum of 42 (forty-two!) review papers have been cited. Unfortunately, these are not indicated as such, and authors make it look like these are primary citations. This should be addressed and if a review is cited, authors should indicate this. For example: as reviewed by Abbott et al, 2006 (ref number). Some references should be checked if they are properly listed (e.g Gundert-Remy 2010) or for duplications, such as Schurhoff and Toborek 2023, Ding et al 2010. Moreover, in some cases it is important to indicate whether a specific study has been done in vitro or in vivo (e.g. existence of tunneling nanotubes).
- Thank your again for the comments, we have checked the references to make sure that they are listed properly and no duplication. We have added the “as reviewed by ……” if applicable. We have also added in vitro or in vivo for specific studies.

Reviewer 2 Report
Comments and Suggestions for Authors
The authors integrate historical perspectives, cellular elements, regulatory mechanisms, and pathological implications and reviewed the BBB in detail. These are very interesting that pericytes have immunomodulatory properties, that pericytes and astrocytes communicate through tunneling nanotubes in response to astrocyte apoptosis triggered by ischemia or reperfusion injury, and that pericytes have the ability of differentiate into other cell types, including endothelial cells. As this review paper covers important characteristics of BBB, the paper should be accepted for publication in Cells. On the other hand, it would be nice to review findings on intramural periarterial drainage (IPAD) pathway and glymphatic system, as two pathways have important roles in maintenance of the stability of the central nervous system.
Author Response
Dear editors and reviewers,
We would like to thank the editors for giving us a chance to resubmit this manuscript, and also thank the reviewers for giving us constructive suggestions on our manuscript entitled “Recapitulation on the structure-function-regulation of the Blood-Brain Barrier in (patho)physiological conditions”. Those comments are all valuable and useful in helping us to revise and improve our paper. We have considered the comments carefully and have made a point-to-point response and we hope you reconsider this revised manuscript for publication.
Sincerely,
Yuechun Wang
Reviewer #2:
Thank you for the comments. It is really good suggestions to review findings on intramural periarterial drainage (PAD) pathway and glymphatic system which play important roles in maintenance of the stability of the central nervous system. Due to the limitations of the paper, we just mentioned the PAD pathway and glymphatic system briefly when introduced the homeostasis of CNS in the section of introduction.

Reviewer 3 Report
Comments and Suggestions for Authors
In this review of scientific data concerning the blood-brain barrier, the authors have focused their analysis on structure-function-regulation relationships in physiological and pathophysiological conditions by showing the interest and complementary role played by the different cell types composing the BBB.
Starting with scientific history, the authors have clearly highlighted the evolution of this knowledge and the molecular as well as cellular links explaining BBB function and regulation. The tabular summaries are highly appreciated and make it easier to understand the pathological processes and their role in BBB permeability.
Unfortunately, the authors have included only one bibliographical reference for each line of the table, whereas a much larger number of references exist.
Furthermore, the quality of the images is very poor (fig 1), making the written indications illegible.
Author Response
Dear editors and reviewers,
We would like to thank the editors for giving us a chance to resubmit this manuscript, and also thank the reviewers for giving us constructive suggestions on our manuscript entitled “Recapitulation on the structure-function-regulation of the Blood-Brain Barrier in (patho)physiological conditions”. Those comments are all valuable and useful in helping us to revise and improve our paper. We have considered the comments carefully and have made a point-to-point response and we hope you reconsider this revised manuscript for publication.
Sincerely,
Yuechun Wang
We greatly appreciate your comments. We have updated the manuscript to include more references that better represent the breadth of the topic.
Additionally, we have improved the quality of Figure 1. to enhance the visual clarity of the images, ensuring that the written indications are now clearly legible.

Round 2
Reviewer 1 Report
Comments and Suggestions for Authors
The authors have responded largely to the specific comments, but still fall short to make this a review that adds anything significant to the existing literature. Still a significant portion of this manuscript is a rehash of existing reviews on the BBB. English grammar, punctuation and spacing still needs attention and running a word check may help as well. As authors try to cover so many topics, none of them ends up well covered. The suggestion is to be more selective and cover specific topics more thoroughly and in depth, e.g. the role of heavy metals, which seem to have the author’s interest. Authors should really improve the manuscript beyond the provided comments, as there are way too many issues. Still, too many review papers are referenced.
Note that when a review paper is referenced, it should be indicated as such (see review by Zlokovic et al, 3) and not just ‘3’. More original primary and recent ‘up-to-date” references should be included and not predominantly older reviews. Still an excessive number of review papers are referenced. For example, section 3.4.1 on circadian rhythm, contains 2 review papers out of 3 and the third pertains to the gut and not BBB, and this is not made clear. Section 3.4.2 out of 8 references, half are reviews. In section 3.4.4, last sentence, here the authors correctly referred to a review.
3.4.5.: on brain EC (not in) and brain (?) parenchyma.
The authors have now included a list of abbreviations, but ignored the suggestion that abbreviations are not needed if the term is only used a few times and not throughout the whole manuscript.
The background and history of the discovery of the BBB still has inconsistencies in it and also needs proper referencing. Goldman was not a colleague of Ehrlich but his student (who ‘outsmarted’ his boss by injecting into the brain). Lena Stern effectively developed the concept of the BBB as a selective and semipermeable barrier, not just bidirectional, which is a very different concept from semipermeable. Its impressive that the Lena Sterns paper (1922) in French was references, but maybe the some points got lost in the translation. The site of the endothelium as BBB was not identified till Karnosvky and not Biedl and Kraus. It is also too long wordy and includes some repetitive unnecessary long sentences. This section can be significantly shortened. Importantly, this will free up the needed space for the authors to expand on the more novel and more interesting section of the manuscript on the effect of heavy metals on the BBB, which currently falls short (and as previously suggested). Currently, they state that they cannot expand on it due to the word limitations.
Again, it does not seem completely clear that the authors see the BBB is a part of the NVU. Also there is not a triple barrier, but there are three sites of the barrier, which is a fundamentally different concept.
Section 4.1 , authors should clearly include microbes (neurotropic microbes) and not generic biological factors
Page 12, “here we take heavy metals….” Sentence is just standing by itself and should be included in 4.1.1. This paragraph can also be expanded, with potential mechanisms (al sections of 4.1.x). As is, it is not informative. Authors referring here to three (!) reviews: 103,-104, 105. Again, were referring to a review, it should be indicated as such. Etc etc throughout the manuscript.
Thre are a lot of unclear and not (properly) referenced statements all over the manuscript. Another example (and I cant give all, there are too many) section 4.2.1: endothelial cells “thrombin allows birth of endothelial cells and astrocytes”. How should we envision this?
Tope page 15: recombinant not R
Also is fibriblast growth factors all have different actions, discrimination should be made. Besides an angiogenesis promoting factor does not upregulate TJ’s, but increases permeability.
TNT’s is only used here and does not need abbreviation, as stated earlier.
Ding is still twice referenced as 124 and 130
Comments on the Quality of English Language
still authors need to check punctuation, and spacing.
A word checker is also recommended
Author Response
Dear Reviewer,
Thank you for your thorough and insightful feedback on our paper. We appreciate your time and expertise in helping us improve our manuscript. We have addressed each of your points as follows.
We hope that these responses address your concerns effectively and improve the quality of our manuscript. Thank you again for your invaluable feedback.
Best regards,
Hin Fong
Thank you for the comments. We have reviewed the entire manuscript, addressing and correcting all grammar, punctuation, and spacing issues sentence by sentence.
2. Note that when a review paper is referenced, it should be indicated as such (see review by Zlokovic et al, 3) and not just ‘3’.
We appreciate the suggested modifications you provided for the citation style in our paper. Now we have corrected the citation style for review literature, as indicated below.
This complex signaling mechanism ultimately influences BBB integrity by modulating the oscillation of TJPs and transporter functionality, which is demonstrated in the review paper by Nicolette S; Michal T[74].
This cascade culminates in the synthesis of glucocorticoids, as summarized in the review paper by Charmandari E, Tsigos C, Chrousos G [79].
3. More original primary and recent ‘up-to-date” references should be included and not predominantly older reviews. Still an excessive number of review papers are referenced. For example, section 3.4.1 on circadian rhythm, contains 2 review papers out of 3 and the third pertains to the gut and not BBB, and this is not made clear. Section 3.4.2 out of 8 references, half are reviews. In section 3.4.4, last sentence, here the authors correctly referred to a review.
We appreciate your identification of errors in the citation content of this paper. The section on 'gut' has been removed and replaced with other specific example related to the circadian regulation of the BBB.
For example, elevated Mg2+ levels in daytime subperineurial glia, which is maintained by molecular clock in perineurial glia cells, enhance the activity of P-glycoprotein transporters, leading to a reduction in BBB permeability.
And we reduced the citations of review-type literature in favor of increasing original references, especially in sections 3.4.1 and 3.4.2.
4. 3.4.5.: on brain EC (not in) and brain (?) parenchyma.
Thank you for the comment.We have corrected it as below.
Research suggests that heightened neural activity resulting from exercise could elevate brain levels of Insulin-like Growth Factor 1, which is secreted by the liver and binds to Insulin-like Growth Factor 1 receptors that are abundant in CECs.
5. The authors have now included a list of abbreviations, but ignored the suggestion that abbreviations are not needed if the term is only used a few times and not throughout the whole manuscript.
We appreciate your suggestions regarding the abbreviation table. We have streamlined the list, removing words that appear only a few times.
6. The background and history of the discovery of the BBB still has inconsistencies in it and also needs proper referencing.
Thank you for your reminder in the inconsistencies in the background and history of the BBB's discovery and the lack of proper referencing. We have carefully reviewed and revised the manuscript to ensure accuracy and provide appropriate references for the background and history of the BBB.
The BBB has been a topic of intense research interest since its postulation in the late 19th century. Paul Ehrlich, using tissue staining techniques, observed that certain dyes injected into the peripheral circulation permeated almost all tissues but not the brain and CSF. Goldmann, a student of Ehrlich, conducted further research in 1909 and 1913. He found that injecting the dye Trypan blue into the brain ventricles resulted in staining of the brain tissue, while intravenous administration did not stain the brain or spinal cord. He then hypothesized that the CSF, entering the brain tissue via the choroid plexuses, acted as a transporter for certain substances. These observations and hypotheses contributed to the understanding of the BBB and its role in regulating the passage of substances between the bloodstream and the brain.
7. Goldman was not a colleague of Ehrlich but his student (who ‘outsmarted’ his boss by injecting into the brain).
Thank you for pointing out the error. We have made the necessary correction in the revised manuscript to accurately reflect this information:
“Goldmann, a student of Ehrlich found that…”
8. Lena Stern effectively developed the concept of the BBB as a selective and semipermeable barrier, not just bidirectional, which is a very different concept from semipermeable.
We appreciate your clarification regarding Lena Stern's contribution to the concept of the BBB. We have revised the text to accurately reflect that Lena Stern effectively developed the concept of the BBB as a selective and semipermeable barrier, which is indeed a different concept from bidirectional permeability:
“Their findings suggest that the interface between the brain and the rest of the body is semipermeable and operates bidirectionally…”
9. The site of the endothelium as BBB was not identified till Karnosvky and not Biedl and Kraus.
Thank you for pointing out the error in attributing the identification of the endothelium as the BBB to Biedl and Kraus. We have rectified this mistake and accurately credited Karnovsky in the revised manuscripts follows:
“Reese and Karnovsky finally determined that the BBB mainly consists of astrocytic end feet and capillary endothelium in the late 1960s using electron microscopic cytochemical studies.”
10. It is also too long wordy and includes some repetitive unnecessary long sentences. This section can be significantly shortened.
We appreciate your feedback regarding the length and wordiness of the section. We agree that it can be significantly shortened to improve clarity and conciseness. We have carefully revised the text, eliminating repetitive and unnecessary long sentences to make it more concise while still effectively conveying the relevant information.
11. Importantly, this will free up the needed space for the authors to expand on the more novel and more interesting section of the manuscript on the effect of heavy metals on the BBB, which currently falls short (and as previously suggested). Currently, they state that they cannot expand on it due to the word limitations.
Thank you for comment, we understand your suggestion to expand on the section discussing the effect of heavy metals on the BBB, which is a more novel and interesting aspect of the manuscript. We have made efforts to include additional relevant information and findings within the given space to enhance the discussion.
12. Again, it does not seem completely clear that the authors see the BBB is a part of the NVU. Also, there is not a triple barrier, but there are three sites of the barrier, which is a fundamentally different concept.
Thank you for pointing out this confusion caused by our previous statements. We want to clarify that we do recognize the BBB as an integral part of the NVU. We appreciate your clarification that there are three sites of the barrier rather than a triple barrier itself, and we have revised the text to accurately reflect this distinction and corrected as follows:
NVU is a complex functional unit composed of both the nervous component and the vascular component, mainly including neurons, glial cells, endothelial cells, pericytes, and the basement membrane and extracellular matrix.NVU plays an important role in maintaining cerebral vascular function and structural stability. While BBB is an important barrier between blood and brain tissue. Transendothelial fluid transfer, BBB integrity, and neurovascular coupling are all maintained by the collective action of NVU cells.
And “Figure 2. The Blood-Brain Barrier and its neighboring cells The figure illustrates the BBB and its interactions with the neighhoring cells. The BBB is comprised of three distinct sites.”
13. Authors should clearly include microbes (neurotropic microbes) and not generic biological factors authors should clearly include microbes (neurotropic microbes) and not generic biological factors
In response to your comment regarding the inclusion of microbes in our article, we have made significant revisions to highlight the role of neurotropic microbes specifically. We have changed the table as follows:
Infection-related damage |
Neutrophil microbes such as Neisseria meningitides
|
Release of cvtokines and chemokines by immune cells. Disruption of tight junctio Physical distruption Formation of Neutrophil Extracellular Traps |
Meningitis, Encephalitis, Neurosyphilis, Toxoplasmosis, AIDS |
Goverdhan, P., & Pachter, J. S. (2012)[97]. H. S. L. M. (1999) [74] Varatharaj, A., & Galea, I. (2016)[75] |
14. "here we take heavy metals...." Sentence is just standing by itself and should be included in 4.1.1. This paragraph can also be expanded, with potential mechanisms (al sections of 4.1.x). As is, it is not informative.
Thank you for your time and effort in reviewing our manuscript. After carefully considering your feedback, we have decided to retain the sentence as it is. The intention behind this sentence was to introduce the subsequent discussion, where we use the example of heavy metals to illustrate how they can damage the BBB. We believe that by keeping the sentence as a standalone statement, it effectively serves as a clear transition into the subsequent section.
In response to this comment, we have expanded the paragraph in section 4.1.1 to provide a more thorough discussion of the topic, the changes is as follows:
4.1.1 Heavy metals such as lead, mercury, cadmium, and arsenic can directly target the proteins involved in forming tight junctions between ECs in the BBB[114-116]. As reviewed by Kim J-H[114-116]. These metals can disrupt the structure and function of TJPs, leading to a breakdown in the integrity of the BBB.
Zona occludens-2, a critical component of the zona occludens protein family, plays a pivotal role in the structural organization of TJs across epithelial and endothelial cell layers. These proteins are essential for anchoring the transmembrane protein complexes to the cytoskeleton made of actin, thereby facilitating the proper placement and stabilization of intercellular junctions. The integrity of BBB is partly reliant on the proper functioning of zona occludens-2[117]. Research findings have shown that when exposed to lead, there is a significant reduction in the concentration of zonaoccludens-2, with levels diminishing by 25%. This suggests that lead exposure directly undermines the TJ's integrity.
15. Authors referring here to three (!) reviews: 103,-104, 105. Again, were referring to a review, it should be indicated as such.
Thank you for your comment, we have already added “as reviewed by”
As reviewed by Kim J-H, , these metals can disrupt the structure and function of TJPs……
16. Etc etc throughout the manuscript.
Thank you for your comment. But in the entire text we just identified a total of 1 occurrence of etc. Considering "such as" and "etc" have the similar meaning,we have deleted the “etc”.
17. section 4.2.1: endothelial cells "thrombin allows birth of endothelial cells and astrocytes".
' How should we envision this?
Thank you for the question. Regarding your query about how to envision the role of thrombin in section 4.2.1, specifically its effect on the birth of endothelial cells and astrocytes, we would like to provide a detailed explanation as follows:
In the case of intracerebral hemorrhage, thrombin is clinically used. When injected, it injures the CECs and astrocytes which are are critical for maintaining the functions of the BBB. But after the initial injury, the brain initiates a repair process. New brain microvascular ECs and astrocytes start to form around the damaged vessels. This is the body's natural response to heal and restore the integrity of the BBB.
18. recombinant not R
Thank you! We have changed it already.
19. Also is fibroblast growth factors all have different actions, discrimination should be made.
Thank you for your comment, it should be fibroblast growth factors 20.
20.Besides an angiogenesis promoting factor does not upregulate TJ's, but increases permeability.
Thank you for the comment. After reading the origin article carefully, we have changed it as below.
According to the study by Guo et al. (2021b), during the cute stage of traumatic brain injury, recombinant human fibroblast growth factor 20 could upregulate the expression of TJ proteins and adherens junction proteins, and also promote angiogenesis, so alleviated BBB damage.
21. TNT's is only used here and does not need abbreviation, as stated earlier.
Thank you! We have changed it with the full name of tunneling nanotubes.